# Male sex determination maintains proteostasis and extends lifespan of *daf-18/PTEN* deficient *C. elegans*

Zhi Qu[1,2,5], Lu Zhang[3,5], Xue Yin[3,5], Fangzhou Dai[3], Wei Huang[3], Yutong Zhang[3], Dongyang Ran[3] & Shanqing Zheng [ID][1,3,4✉]

## Abstract

**Although females typically have a survival advantage, those with PTEN functional abnormalities face a higher risk of developing tumors than males. However, the differences in how each sex responds to PTEN dysfunction have rarely been studied. We use *Caenorhabditis elegans* to investigate how male and hermaphrodite worms respond to dysfunction of the PTEN homolog *daf-18*. Our study reveals that male worms can counterbalance the negative effects of *daf-18* deficiency, resulting in longer adult lifespan. The survival advantage depends on the loss of DAF-18 protein phosphatase activity, while its lipid phosphatase activity is dispensable. The deficiency in DAF-18 protein phosphatase activity leads to the failure of dephosphorylation of the endoplasmic reticulum membrane protein C18E9.2/SEC62, causing increased levels of unfolded and aggregated proteins in hermaphrodites. In contrast, males maintain proteostasis through a UNC-23/NEF-mediated protein ubiquitination and degradation process, providing them with a survival advantage. We find that sex determination is a key factor in regulating the differential expression of *unc-23* between sexes in response to *daf-18* loss. These findings highlight the unique role of the male sex determination pathway in regulating protein degradation.**

**Keywords** *C. elegans*; *daf-18*; Male Sex Determination; Longevity; *unc-23*
**Subject Categories** Development; Molecular Biology of Disease; Post-translational Modifications & Proteolysis

## Introduction

Phosphatase and tensin homolog on human chromosome 10q23.3 (*PTEN*) was classified as a potential tumor suppressor (Steck et al, 1997). PTEN dysfunction causes autosomal dominant *PTEN* hamartoma tumor syndrome (PHTS) (Pilarski et al, 2013). Autosomal *PTEN* mutations can be genetically inherited by both sexes; however, there is a strong predominance of PTEN-deficient females, who have a high incidence of developing breast and endometrial cancers during aging (Starink et al, 1986). Sex-specific differences are widely reported to be found in regulating tumor incidence and mortality (Rubin, 2022). Unfortunately, how different sexes specifically respond to the cancer risk and reduced survival caused by dysfunction of certain autosomal tumor suppressors has been underexplored.

We used *Caenorhabditis elegans* (*C. elegans*) as a model to study the sex-specific difference in the response to loss of *daf-18* (the *PTEN* ortholog in *C. elegans*). We found that males survived better than hermaphrodites when *daf-18* was lost. Same as PTEN, DAF-18 is a dual phosphatase containing lipid and protein phosphatase activities. The well-known DAF-18 lipid phosphatase can dephosphorylate phosphatidylinositol-3, 4, 5-trisphosphate (PIP3) to phosphatidylinositol-4, 5-bisphosphate (PIP2), thereby reducing the activity of insulin/insulin-like signaling pathway (IIS) in *C. elegans* (Ogg and Ruvkun, 1998). IIS is widely recognized as a pivotal pathway that controls aging and stress resistance (Hsu et al, 2003). We found that IIS was excluded from controlling the sex bias of survival when *daf-18* is lost, and the loss of transcription factors regulated by IIS did not confer males a survival advantage.

In *C. elegans*, individuals with two X chromosomes (XX) and two sets of autosomes develop a female soma and a hermaphroditic germline, whereas individuals with one X chromosome (XO) and two sets of autosomes develop male somatic and germline features. Both somatic and germline sexual fates are governed by the same core cascade, consisting of several genes in the global sex determination pathway (Goodwin and Ellis, 2002). This global sex determination pathway consists of seven essential genes, including *her-1*, *tra-2*, *tra-3*, and *fem-1*, *fem-2*, and *fem-3*, which collectively control sexual identity in both somatic and germ cells (Goodwin and Ellis, 2002; Hodgkin, 1987). The *her-1* gene directs male development by promoting male-specific somatic features and continuous spermatogenesis. HER-1 achieves this by binding to and inactivating the transmembrane protein TRA-2, which otherwise promotes female development (Kuwabara and Kimble, 1995). TRA-2, in turn, requires TRA-3 to support female differentiation (Barnes and Hodgkin, 1996a). The FEM proteins (FEM-1, FEM-2, and FEM-3) also play a crucial role in male

[1]The Zhongzhou Laboratory for Integrative Biology, Henan University, 450000 Zhengzhou, Henan, China. [2]School of Nursing and Health, Henan University, 475004 Kaifeng, China. [3]School of Basic Medical Sciences, Henan University, 475004 Kaifeng, China. [4]Laboratory of Cell Signal Transduction, Henan Provincial Engineering Centre for Tumor Molecular Medicine, Medical School of Henan University, 475004 Kaifeng, China. [5]These authors contributed equally: Zhi Qu, Lu Zhang, Xue Yin. ✉E-mail: zhengshanqing@henu.edu.cn

development, acting downstream of TRA-2 to regulate somatic sex (Chin-Sang and Spence, 1996; Hodgkin, 1986). The FEM proteins function in a negative regulatory hierarchy to modulate TRA-1 activity, a central but not ultimate regulator of sexual fate (Hodgkin, 1987). However, despite their importance, the specific molecular targets of the FEM proteins remain largely unknown. This gap in understanding of FEM function and downstream regulatory interactions presents a significant area for ongoing research in sex determination and sex bias survival regulation. The underlying causes of sex differences in survival and aging remain mostly unknown. The possible reasons are currently speculated to be genome instability (Bronikowski et al, 2022), sex chromosome dosage loss (Sano et al, 2022), mutation buffering effects (Rubin, 2022), and sex hormone- and receptor-regulated responses (Hickey et al, 2021). However, sex-specific responses to autosomal gene deficiency are basically unreported.

In this study, we observed that loss of *daf-18* led to a heightened unfolded protein response in the endoplasmic reticulum (ER) and global protein aggregation in hermaphrodites. Interestingly, males exhibited a specific activation of the ubiquitin-proteasome system (UPS) to mitigate this stress, primarily by inducing UNC-23 to enhance protein degradation and maintain a survival advantage. *daf-18* males or hermaphrodites with enhanced global sex determination, achieved by manipulating the central genes including *her-1*, *tra-2* and *fem-2* in the sex determination cascade, have the potential to restore protein homeostasis by regulating the expression of *unc-23*. This may elucidate how males maintain a survival advantage in the absence of *daf-18*. Considering that the sex determination cascade governs sexual differences and is initiated in the early embryo, our findings could fundamentally explain why different sexes respond differently to *daf-18* loss. Moreover, this insight may offer a novel research and therapeutic perspective for sex-biased diseases.

## Results

### Males outlive hermaphrodites when *daf-18* is lost

Survival of L1-arrested *daf-18(ok480)* hermaphrodites is drastically reduced (from 21 days to 4 days) (Zheng et al, 2018a). When *daf-18(ok480);him-5(e1490)* mutants were tested for survival during L1 arrest, we found that some L1-arrested worms lived longer than 4 days (Fig. EV1A). These surviving worms were recovered from L1 arrest and cultured on NGM plates with food. We found that they were all male worms, suggesting that *daf-18(ok480)* males live longer than *daf-18(ok480)* hermaphrodites during L1 arrest. To more precisely analyze the survival of *daf-18*-deficient worms in L1 arrest, *daf-18(ok480);him-5(e1490)* worms were cultured in M9 plus 0.08% (v/v) ethanol, which can increase the survival of in L1-arrested worms (Fukuyama et al, 2015) (Fig. EV1B). We found that the percentages of surviving hermaphrodites decreased over time during L1 arrest (Fig. 1A); interestingly, however, the percentages of surviving L1 arrested males increased over time (Fig. 1B). Next, we tested the adult lifespan of males and hermaphrodites under normal feeding. We found that the adult lifespans of hermaphrodites of *daf-18(ok480);him-5(e1490)* worms were reduced significantly by loss of *daf-18* (Fig. 1C); in contrast, the adult lifespans of *daf-18(ok480);him-5(e1490)* males were actually extended by loss of

*daf-18* (Fig. 1D). In humans and nematodes, females or hermaphrodites live longer than males. We tested the adult lifespan of *him-5(e1490)*, and our results confirmed that the hermaphrodites lived longer than the males (Fig. 1E). However, when *daf-18* was lost, the adult lifespans of males were significantly longer than those of hermaphrodites (Fig. 1F).

We utilized the *him-5(e1490)* mutation to generate large-scale male progenies. Although the control worms also had the *him-5(e1490)* mutation during the adult lifespan analysis, we repeated the key adult lifespan experiments using worms without *him-5(e1490)* to further exclude any potential effects of *him-5(e1490)* on sex-biased survival. This was necessary because *him-5(e1490)* is involved in X chromosome dynamics (Broverman and Meneely, 1994). Our results demonstrated that the male survival advantage is related to *daf-18*, not *him-5* (Appendix Fig. S1 and Appendix Table S1). The *ok480* mutation is a deletion in the *daf-18* gene that has been widely reported to cause a severe loss of *daf-18* activity (Zheng et al, 2018b). However, in order to verify whether the observed male survival advantage is a characteristic of the *ok480* allele alone, additional *daf-18* alleles (*mg198*, *nr2037* and *e1375*) were tested in adult lifespan experiments involving both males and hermaphrodites. The results support our findings and confirm the male survival advantage in all tested *daf-18* mutants (Appendix Fig. S2 and Appendix Table S2).

Some studies have also suggested that males live longer than hermaphrodites when cultured individually to avoid interactions (Gems and Riddle, 2000). Although, when cultured individually (1 worm in 1 plate), *him-5(e1490)* and wild-type males live longer than hermaphrodites, hermaphrodites exhibit a significantly reduced adult lifespan when the *daf-18* gene is lost (Appendix Fig. S3A and Appendix Table S3). In contrast, *daf-18(ok480)* males still possess a survival advantage and even live longer than control worms (Appendix Fig. S3B and Appendix Table S3).

Taken together, these results show that males have the potential to compensate for *daf-18* loss, allowing the males survive better than hermaphrodites, and suggest that there are sex-specific differences in the response to *daf-18* loss.

### *unc-23* promotes male survival when *daf-18* is lost

As one single autosomal gene dysfunction caused males and hermaphrodites to respond differently, the different gene expression signatures were compared within each sex, males (with or without *daf-18 ok480*) and hermaphrodites (with or without *daf-18 ok480*), by integrating RNA sequencing (RNA-seq) transcriptomic data. The transcriptional consequences of *daf-18* loss in hermaphrodites and males were investigated. The gene expression signature changes in males were far more enormous than those in hermaphrodites. The *daf-18(ok480)* males had more specific genes that were downregulated or upregulated by *daf-18* loss (Fig. 2A). Considering that the male response to *daf-18* loss was opposite to that of hermaphrodites, we focused on the genes whose expression changed inversely in males and hermaphrodites (Fig. 2B). In total, 131 genes were downregulated by more than twofold in males and upregulated in hermaphrodites, while 67 other genes were upregulated by more than 2-fold in males and downregulated in hermaphrodites (Fig. 2B). All 198 genes were used to perform Kyoto Encyclopedia of Genes and Genomes (KEGG) pathway analysis. The data showed that these genes were primarily enriched

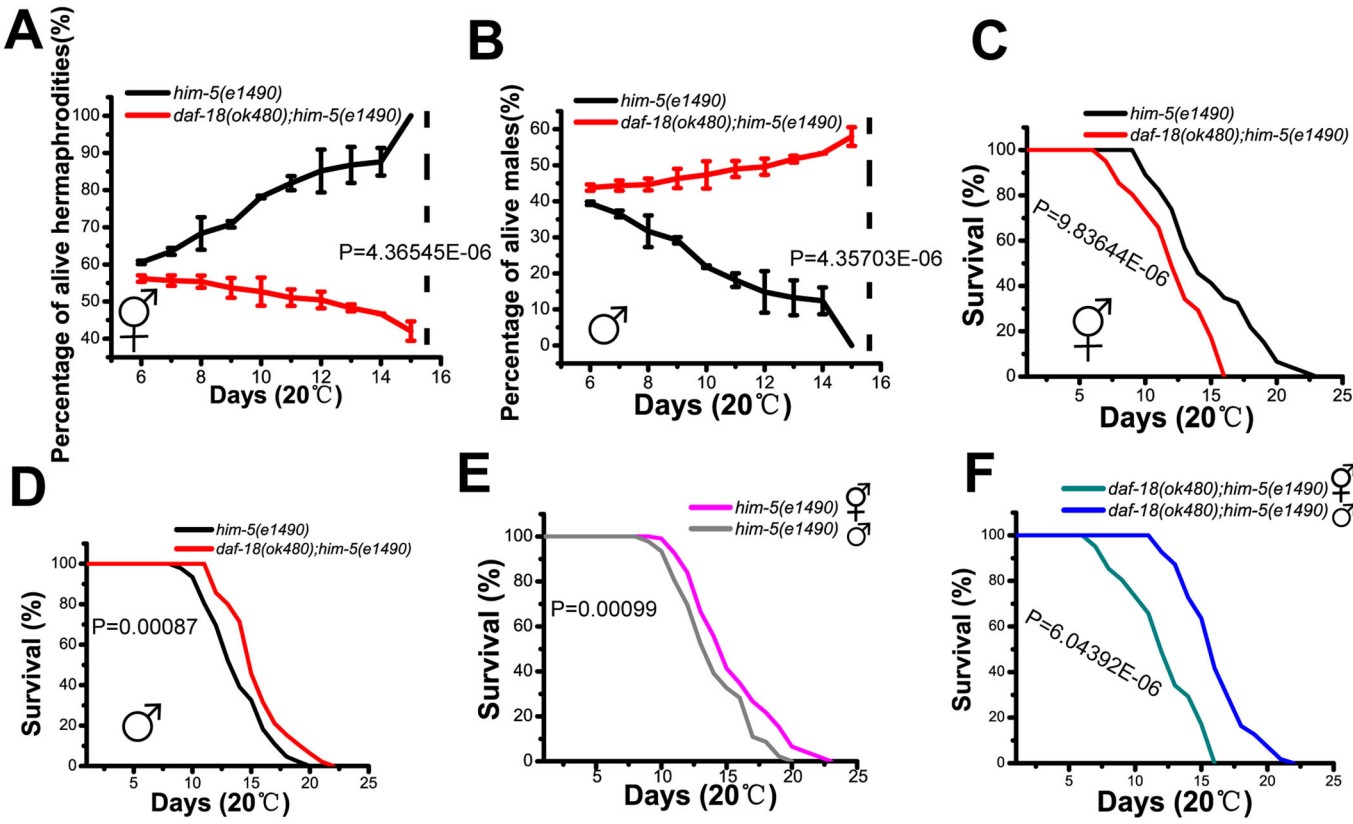

**Figure 1. Males survive better than hermaphrodites when *daf-18* is lost.**

(A) Percentages of live hermaphrodites recovered from L1 arrest each day. The data show the average of three independent repeats, and the error bars show the standard deviations. *P* values were determined by using a two-tailed *t* test. (*P* value; *him-5(e1490)* hermaphrodites vs *daf-18(ok480);him-5(e1490)* hermaphrodites: 4.36545E-06). (B) Percentages of live males recovered from L1 arrest each day. The data show the average of three independent repeats, and the error bars show the standard deviations. *P* values were determined by using a two-tailed *t* test. (*P* value; *him-5(e1490)* males vs *daf-18(ok480);him-5(e1490)* males: 4.35703E-06). Every day, the L1 arrest worms were collected and the proportion of males in the population of living worms was determined. *him-5(e1490)* sample size (*n*) = 377 (day 6), 346 (day 7), 407 (day 8), 432 (day 9), 426 (day 10), 433 (day 11), 437 (day 12), 280 (day 13), 192 (day 14), 98 (day 15). *daf-18(ok480);him-5(e1490)* sample size (*n*) = 530 (day 6), 726 (day 7), 327 (day 8), 342 (day 9), 296 (day 10), 273 (day 11), 251 (day 12), 215 (day 13), 170 (day 14), 123 (day 15). Survival curves of adult hermaphrodites (C) and males (D) when *daf-18* was lost. The *him-5(e1490)* mutation was used to generate male progeny. (E) Difference in survival between adult hermaphrodites and males of *him-5(e1490)* worms. (F) Difference in survival between adult hermaphrodites and males of *daf-18(ok480);him-5(e1490)* worms. *him-5(e1490)* hermaphrodite sample size (*n*) = 92, *him-5(e1490)* male sample size (*n*) = 77, *daf-18(ok480);him-5(e1490)* hermaphrodite sample size (*n*) = 96, *daf-18(ok480);him-5(e1490)* male sample size (*n*) = 101. (C–F) Each survival curve is representative of three independent repeats. *P* values were determined by using the log rank test (*P* values; C, *him-5(e1490)* hermaphrodites vs *daf-18(ok480);him-5(e1490)* hermaphrodites: 9.83644E-06; D, *him-5(e1490)* males vs *daf-18(ok480);him-5(e1490)* males: 0.00087; E, *him-5(e1490)* hermaphrodites vs *him-5(e1490)* males: 0.00099; F, *daf-18(ok480);him-5(e1490)* hermaphrodites vs *daf-18(ok480);him-5(e1490)* males: 6.04392E-06). Source data are available online for this figure.

in the ER-related protein processing pathway (Fig. 2C). The RNA-Seq data showed that the expression of five genes related to this pathway changed in opposite directions in males and hermaphrodites (Fig. 2D), and the expression profiles of these genes were further confirmed by using real-time PCR (Fig. 2E). The expression of these genes was tested in wild-type males and hermaphrodites. We found that the expression of these genes did not differ between sexes (Appendix Fig. S4), suggesting that the change in gene expression is specific to *daf-18* loss. The effects of all these genes on the survival of male and hermaphroditic *daf-18(ok480)* mutants were tested. We found that *pek-1*, *skr-8* and *F44E5.4/5* RNAi failed to change the survival of *daf-18(ok480)* males or hermaphrodites, while knocking down *unc-23*, which was upregulated in *daf-18(ok480)* males, caused the percentage of surviving males to drop significantly over time during L1 arrest (Fig. 2F). We also

overexpressed *unc-23* in *daf-18(ok480)* hermaphrodites and found that the decrease in the percentage of surviving L1-arrested hermaphrodites due to *daf-18* loss was rescued (Fig. 2G). Next, we tested the effects of changing *unc-23* expression on adult lifespan. Overexpression of *unc-23* in *daf-18(ok480)* hermaphrodites increased adult lifespan (Fig. 2H), which was also confirmed by individually culturing lifespan experiment (Appendix Fig. S5A and Appendix Table S3). Knocking down *unc-23* in *daf-18(ok480)* hermaphrodites did not change adult lifespan (Fig. 2I), but knocking down *unc-23* in *daf-18(ok480)* males significantly reduced adult lifespan (Fig. 2J), which was also confirmed by individually culturing lifespan experiment (Appendix Fig. S5B and Appendix Table S3). Furthermore, we compared the adult lifespans of *daf-18(ok480)* males and hermaphrodites when *unc-23* was knocked down and found that the sex-specific difference in the response to

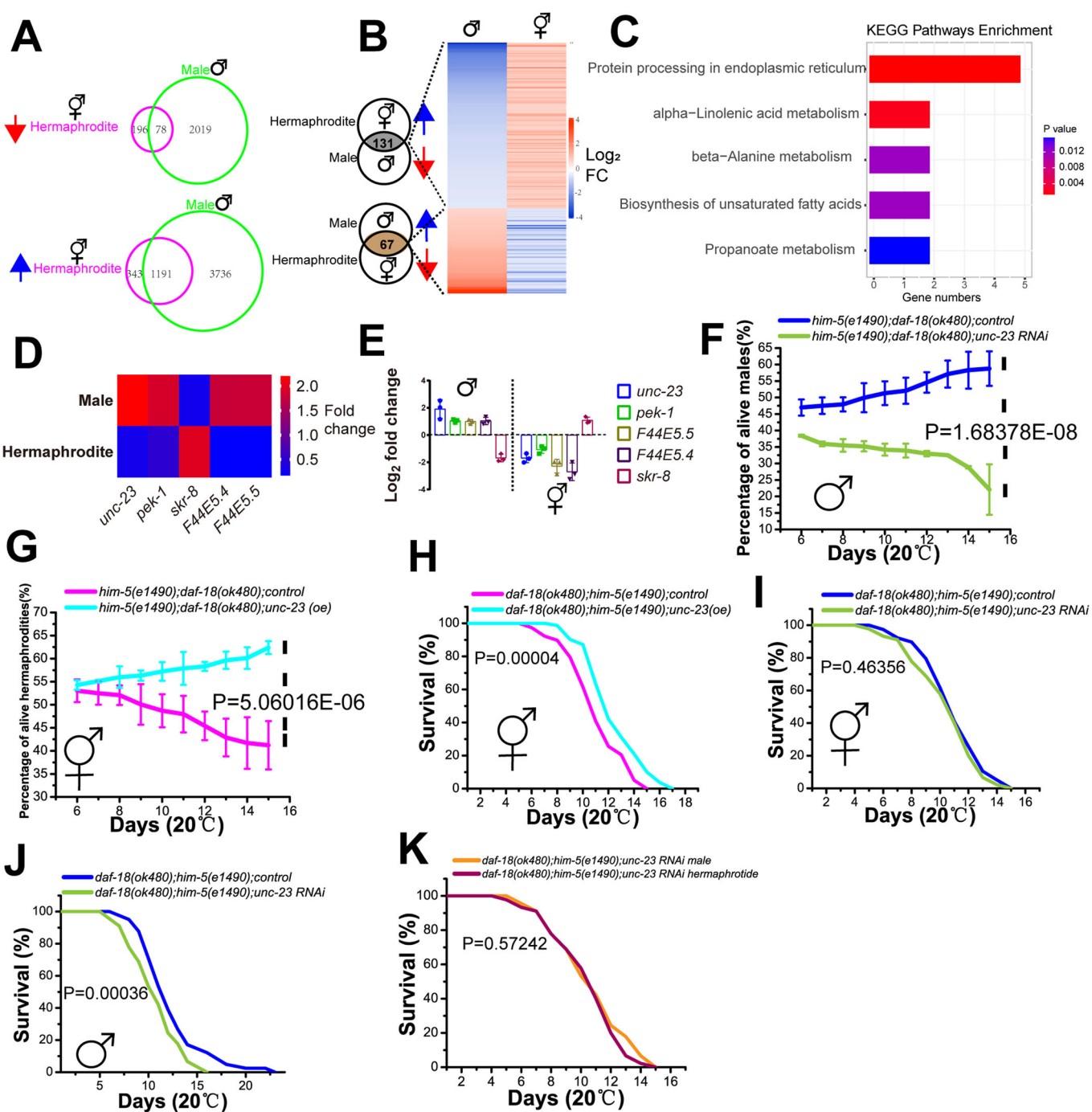

*daf-18* loss disappeared (Fig. 2K). Together, these results show that the sex-specific difference in survival between males and hermaphrodites with *daf-18* loss is achieved through opposite *unc-23* expression profiles in *daf-18(ok480)* males and hermaphrodites.

## *daf-18*-deficient males have elevated protein ubiquitination levels and reduced aggregated protein levels

*unc-23* has been reported to be required for proper attachment of the body wall muscle to the hypodermis (Plenefisch et al, 2000). We thus examined the progression of detachment of the body wall

muscle from the hypodermis in *daf-18(ok480)* males and hermaphrodites by using a muscle GFP marker (*myo-3p*::GFP) (Benedetti et al, 2006). Our results showed that *daf-18(ok480)* hermaphrodites exhibited reduced muscle attachment from the hypodermis, while *daf-18(ok480)* males did not exhibit a significantly altered status of muscle attachment from the hypodermis (Fig. EV2). This result suggests that *unc-23*, which is upregulated in males, helps them to attach their muscles more efficiently than hermaphrodites. UNC-23 is predicted to be an ortholog of human BAG co-chaperone 2 (BAG2)-type nucleotide exchange factor (NEF) (Rahmani et al, 2015), which plays a crucial role in regulating the degradation of unfolded proteins (Rosenzweig et al, 2019). UNC-23 can control

◀

**Figure 2. ER-related protein processing affects the survival of *daf-18*-deficient worms.**

(A) Comparison of the gene expression changes in hermaphrodites and males when *daf-18* is lost. The genes with expression changes more than two folds were selected according to RNA-seq. (B) Inversely expressed genes in *daf-18(ok480)* hermaphrodites and males. Mean log2FC: the average log2(fold change) over three iterations of RNA-seq ( >1). (C) KEGG pathway enrichment of inversely expressed genes in *daf-18(ok480)* hermaphrodites and males. (D) Average fold change in ER-related protein processing gene expression according to RNA-seq. (E) The average log2(fold change) in ER-related protein processing gene expression was tested by using real-time PCR. All these genes were significantly up- or downregulated by at least twofold. The data show the average of three independent repeats, and the error bars show the standard deviations. (F) Knocking down *unc-23* reduced the percentages of surviving L1-arrested *daf-18(ok480)* males. Control: RNAi control clones containing the empty vector L4440. *him-5(e1490);daf-18(ok480);control* sample size (*n*) = 146 (day 6), 167 (day 7), 193 (day 8), 238 (day 9), 132 (day 10), 142 (day 11), 131 (day 12), 179 (day 13), 167 (day 14), 203 (day 15). *him-5(e1490);daf-18(ok480);unc-23* RNAi sample size (*n*) = 156 (day 6), 139 (day 7), 190 (day 8), 152 (day 9), 180 (day 10), 179 (day 11), 122 (day 12), 178 (day 13), 162 (day 14), 132 (day 15). The data show the average of three independent repeats, and the error bars show the standard deviations. The *P* value was determined by using a two-tailed *t* test (*P* value; *him-5(e1490);daf-18(ok480);control* males vs *him-5(e1490);daf-18(ok480);unc-23* RNAi males: 1.68378E-08). (G) Overexpression of *unc-23* improved the survival of L1-arrested *daf-18(ok480)* hermaphrodites. Control: transgenic injection strains with the empty expression vector L2528. *him-5(e1490);daf-18(ok480);control* sample size (*n*) = 146 (day 6), 180 (day 7), 190 (day 8), 225 (day 9), 142 (day 10), 169 (day 11), 155 (day 12), 196 (day 13), 138 (day 14), 219 (day 15). *him-5(e1490);daf-18(ok480);unc-23(oe)* sample size (*n*) = 187 (day 6), 170 (day 7), 175 (day 8), 174 (day 9), 159 (day 10), 145 (day 11), 154 (day 12), 164 (day 13), 175 (day 14), 156 (day 15). The data show the average of three independent repeats, and the error bars show the standard deviations. The *P* value was determined by using a two-tailed *t* test (*P* value; *him-5(e1490);daf-18(ok480);control* hermaphrodites vs *him-5(e1490);daf-18(ok480);unc-23(oe)* hermaphrodites: 5.06016E-06). (H) Overexpression of *unc-23* improved the survival of adult *daf-18(ok480)* hermaphrodites. Control: transgenic injection strains with the empty expression vector L2528. Control sample size (*n*) = 104, *unc-23* (oe) sample size (*n*) = 74. (I) Knocking down *unc-23* did not significantly change the survival of adult *daf-18(ok480)* hermaphrodites. Control: RNAi control clones containing the empty vector L4440. Control sample size (*n*) = 96, *unc-23* RNAi sample size (*n*) = 96. (J) Knocking down *unc-23* decreased the survival of adult *daf-18(ok480)* males. Control: RNAi control clones containing the empty vector L4440. Control sample size (*n*) = 82, *unc-23* RNAi sample size (*n*) = 90. (K) Comparison of the survival of adult hermaphrodites and males of *daf-18(ok480); unc-23* RNAi. Control sample size (*n*) = 92, *unc-23* RNAi sample size (*n*) = 96. Each lifespan experiment set was independently repeated three times. The mean survival rates were calculated using the Kaplan–Meier method, and *P* values were determined by using the log rank test (*P* values; H, *daf-18(ok480);him-5(e1490);control* hermaphrodites vs *unc-23(oe)* hermaphrodites: 0.00004; I, *daf-18(ok480);him-5(e1490);control* hermaphrodites vs *unc-23* RNAi hermaphrodites: 0.46356; J, *daf-18(ok480);him-5(e1490);control* males vs *unc-23* RNAi males: 0.00036; K, *daf-18(ok480);him-5(e1490); unc-23* RNAi males vs hermaphrodites: 0.57242). Source data are available online for this figure.

substrate presentation and release from HSP70s to promote the refolding of misfolded denatured proteins and the degradation of misfolded proteins (Rosenzweig et al, 2019). As *unc-23* and the significantly changed genes in both sexes were enriched in the ER-related protein processing pathway (Fig. 2C), we used the unfolded protein response in ER (UPR[ER]) marker *hsp-4p*::GFP to test the proteostasis in *daf-18(ok480)* worms (Calfon et al, 2002). We found that the *daf-18(ok480)* and *unc-23*-knockdown hermaphrodites had higher UPR[ER] than controls and that knocking down *unc-23* in *daf-18(ok480)* mutants failed to enhance the UPR[ER] (Fig. 3A). Loss of *daf-18* did not change the UPR[ER] condition in males, but knocking down *unc-23* significantly enhanced the UPR[ER] in wild-type and *daf-18(ok480)* males (Fig. 3B). *unc-23* RNAi had an effect on wild-type males more than the hermaphrodites (Fig. 3A,B). To further confirm the results, we tested the unfolded protein levels by using a PROTEOSTAT Protein Aggregation Assay Kit. The results further confirmed that *daf-18(ok480)* and *unc-23*-knockdown hermaphrodites had higher protein aggregation than controls, and that knocking down *unc-23* in *daf-18(ok480)* mutants failed to enhance the protein aggregation (Fig. 3C,D). Loss of *daf-18* did not change the protein aggregation condition in males, but knocking down *unc-23* significantly enhanced the protein aggregation in wild-type and *daf-18(ok480)* males (Fig. 3E,F). We surmised that the aggregated proteins may have been degraded by the NEF-HSP70 protein degradation system in males to support survival. Thus, we next tested the effects of NEF- and HSP70-encoding genes on survival. We found that unlike *unc-23*, *hsp-70* and *hsp-110* had no role in the sex-specific difference in survival in *daf-18(ok480)* worms (Appendix Table S4).

Notably, aging causes a global loss of ubiquitination that can be ameliorated by longevity paradigms (Koyuncu et al, 2021). Ubiquitination-tagged proteins can be recognized and degraded in the proteasome (Vilchez et al, 2014). The denatured unfolded proteins may be degraded by UNC-23/NEF-regulated

ubiquitination (Labbadia and Morimoto, 2015; Rosenzweig et al, 2019). Therefore, we further tested whether protein ubiquitination can be regulated by *daf-18* and *unc-23* in hermaphrodites and males. Our results showed that *daf-18(ok480)* hermaphrodites had lower total ubiquitination (Fig. 3G) and K48-linked ubiquitination (Fig. 3H) than controls and that knocking down *unc-23* did not significantly change the ubiquitinated protein content in *daf-18(ok480)* hermaphrodites (Fig. 3G,H). Interestingly, the *daf-18(ok480)* males had higher total ubiquitination (Fig. 3I) and K48-linked ubiquitination (Fig. 3J) levels than the controls, but knocking down *unc-23* significantly reduced the ubiquitinated protein content in *daf-18(ok480)* males. As *unc-23* was downregulated in *daf-18(ok480)* hermaphrodites, to further confirm that the protein homeostasis in this mutant is regulated by *unc-23*, we also overexpressed *unc-23* in *daf-18(ok480)* hermaphrodites. We found that overexpression of *unc-23* reduced UPR[ER] and protein aggregation levels and enhanced K48-linked ubiquitination (Fig. EV3; Appendix Table S5). Furthermore, unfolded proteins can be degraded by ERAD (endoplasmic reticulum-associated degradation) or secreted into the cytoplasm for degradation by the ubiquitin-proteasome system (Ub-proteasome). Even though *pek-1* is not involved, we investigated whether other UPR[ER] pathways and the ubiquitin ligase complex play a role in this regulation. XBP-1 is required for the UPR[ER] that counteracts cellular stress induced by accumulation of unfolded proteins, *xbp-1* mRNA is induced by ATF6 and spliced by IRE1 in response to ER stress (Yoshida et al, 2001). We found that *xbp-1* is upregulated in *daf-18(ok480)* males (Fig. EV4A), and knocking down *ire-1*, *atf-6*, or *xbp-1* reduces the lifespan of adult *daf-18(ok480)* males (Fig. EV4B); however, it does not abolish the survival advantage as *unc-23* does. SEL-11 functions as an ER E3 ubiquitin ligase that plays a role in UPR[ER] (Sundaram and Greenwald, 1993). Knocking down *sel-11* significantly reduces the survival advantage in adult *daf-18(ok480)* males (Fig. EV4C). This suggests that ERAD is involved in male survival regulation,

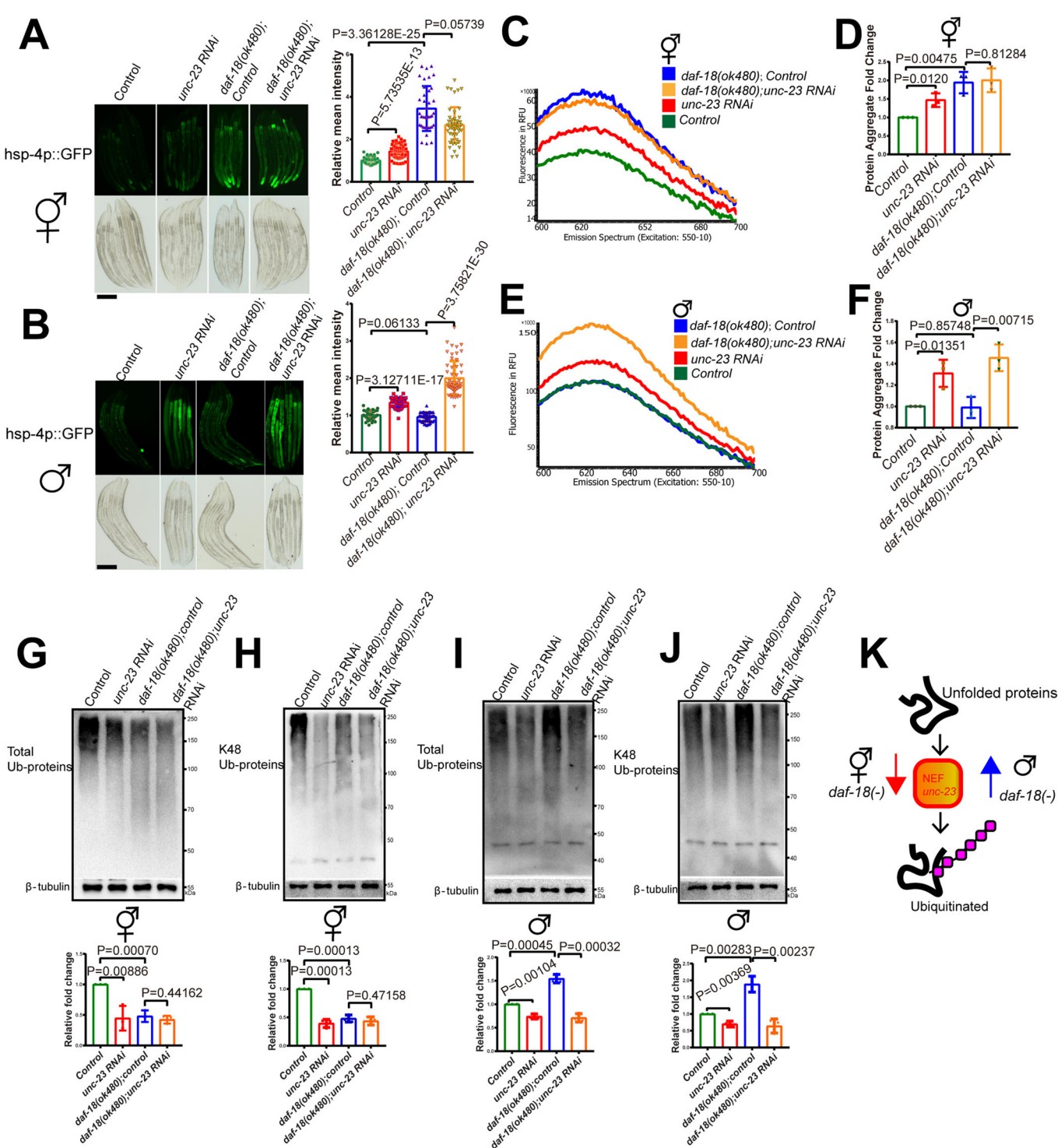

but the UNC-23-regulated Ub-proteasome pathway may play the primary role when *daf-18* is absent.

Taken together, our results show that *daf-18(ok480)* hermaphrodites have increased unfolded protein levels and ubiquitination levels. However, *daf-18(ok480)* males can maintain an elevated ubiquitination level by enhancing protein ubiquitination activity to reduce the unfolded protein status (Fig. 3K), which is regulated by male-specific up-regulation of *unc-23*.

## The lipid phosphatase activity of DAF-18 and IIS do not regulate male survival advantage

The lipid phosphatase activity of DAF-18 works as a negative regulator in the IIS signaling pathway. Loss of *daf-18* inactivates the IIS transcription factors DAF-16 (Paradis and Ruvkun, 1998), HSF-1 (Chiang et al, 2012), and SKN-1 (Tullet et al, 2008), preventing them from achieving positive effects on longevity and stress

Figure 3. *unc-23* regulates ubiquitinated proteins and protein aggregation when *daf-18* is lost.

Unfolded proteins in hermaphrodites (A) and males (B). Hermaphrodite control, *unc-23* RNAi, *daf-18(ok480)* control and *daf-18(ok480);unc-23* RNAi sample size (*n*) = 44, 63, 38 and 50. Male control, *unc-23* RNAi, *daf-18(ok480)* control and *daf-18(ok480);unc-23* RNAi sample size (*n*) = 31, 47, 54 and 55. Each experimental set had three independent repeats. The data shows the values of all samples from three replicates, with error bars representing the averages and standard deviations. The *P* value was determined by using a two-tailed *t* test (*P* values; A, *control* hermaphrodites vs *unc-23* RNAi hermaphrodites: 5.73535E-13; *control* hermaphrodites vs *daf-18(ok480);control* hermaphrodites: 3.36128E-25; *daf-18(ok480);control* hermaphrodites vs *daf-18(ok480);unc-23* RNAi hermaphrodites: 0.05739; B, *control* males vs *unc-23* RNAi males: 3.12711E-17; *control* males vs *daf-18(ok480);control* males: 0.06133; *daf-18(ok480);control* males vs *daf-18(ok480);unc-23* RNAi males: 3.75821E-30). Scale bar: 200 μm. (C) The total level of protein aggregates in hermaphrodites. The data are representative of three independent experiments. (D) Fold changes in protein aggregation in hermaphrodites in three independent experiments. The data show the average of three independent repeats, and the error bars show the standard deviations. The *P* value was determined by using a two-tailed *t* test (*P* values; *control* hermaphrodites vs *unc-23* RNAi hermaphrodites: 0.0120; *control* hermaphrodites vs *daf-18(ok480);control* hermaphrodites: 0.00475; *daf-18(ok480);control* hermaphrodites vs *daf-18(ok480);unc-23* RNAi hermaphrodites: 0.81284). (E) The total level of protein aggregation in males. The data are representative of three independent experiments. (F) Fold changes in protein aggregation in males in three independent experiments. The data show the average of three independent repeats, and the error bars show the standard deviations. The *P* value was determined by using a two-tailed *t* test (*P* values; *control* males vs *unc-23* RNAi males: 0.01351; *control* males vs *daf-18(ok480);control* males: 0.85748; *daf-18(ok480);control* males vs *daf-18(ok480);unc-23* RNAi males: 0.00715). (C–F) Control: RNAi control clones fed the empty vector L4440. (G) Total protein ubiquitination of hermaphrodites. (H) K48-linked protein ubiquitination of hermaphrodites. (I) Total protein ubiquitination of males. (J) K48-linked protein ubiquitination of males. Each immunoblot is representative of three independent experiments. The data show the average of three independent repeats, and the error bars show the standard deviations. The *P* value was determined by using a two-tailed *t* test (*P* values; G, *control* hermaphrodites vs *unc-23* RNAi hermaphrodites: 0.00886; *control* hermaphrodites vs *daf-18(ok480);control* hermaphrodites: 0.00070; *daf-18(ok480);control* hermaphrodites vs *daf-18(ok480);unc-23* RNAi hermaphrodites: 0.44162; H, *control* hermaphrodites vs *unc-23* RNAi hermaphrodites: 0.00013; *control* hermaphrodites vs *daf-18(ok480);control* hermaphrodites: 0.00013; *daf-18(ok480);control* hermaphrodites vs *daf-18(ok480);unc-23* RNAi hermaphrodites: 0.47158; I, *control* males vs *unc-23* RNAi males: 0.00104; *control* males vs *daf-18(ok480);control* males: 0.00045; *daf-18(ok480);control* males vs *daf-18(ok480);unc-23* RNAi males: 0.00032; J, *control* males vs *unc-23* RNAi males: 0.00369; *control* males vs *daf-18(ok480);control* males: 0.00283; *daf-18(ok480);control* males vs *daf-18(ok480);unc-23* RNAi males: 0.00237). (A–J) Control: RNAi control clones containing the empty vector L4440. All the worms contained the *him-5(e1490)* mutation to generate males. (K) Model of the mechanism by which *daf-18*-deficient males maintain protein homeostasis according to our results. When *daf-18* is lost, males can evoke the *unc-23*/NEF-related protein ubiquitination function to keep the unfolded protein level low. Red arrow: the function is decreased. Blue arrow: the function is enhanced. Source data are available online for this figure.

resistance. PQM-1 has also been reported as a target of IIS (Tepper et al, 2013). All these transcription factors have been shown to regulate proteostasis. However, according to our RNA-seq data, the targets of the IIS pathway were similarly affected by the loss of *daf-18* in both males and hermaphrodites. This suggests that the genes controlling the sex bias in survival are not likely to be part of the IIS pathway upon the loss of *daf-18*. To further confirm this, we tested and compared the adult lifespans of males and hermaphrodites with *daf-16(mu86)* (Fig. 4A), *skn-1(zu67)* (Fig. 4B), *pqm-1(ok485)* (Fig. 4C), and *hsf-1(sy441)* (Fig. 4D) mutants. Our results showed that hermaphrodites of these mutants live longer than males, similar to *him-5(e1490)* worms. This suggests that the survival advantage of *daf-18(ok480)* males is not dependent on these transcription factors or the IIS pathway. However, overexpression of full-length *daf-18* under its endogenous promoter can completely abolish the survival advantage of *daf-18* mutant males. (Fig. 4E), suggesting that the survival advantage of males upon loss of *daf-18* may be regulated by other functions of DAF-18. Both DAF-18 and the human tumor suppressor PTEN possess lipid and protein phosphatase activities (Tu et al, 2020). The protein phosphatase activity of DAF-18/PTEN has also been reported to have roles in controlling oncogenic signaling independently of PI3K-AKT (Shi et al, 2014; Shinde and Maddika, 2016; Wozniak et al, 2017). To test the role of the protein phosphatase of DAF-18 in regulating the male survival advantage, we used specific phosphatase-deficient plasmids to rescue the longevity of *daf-18(ok480)* males. Our results showed that *daf-18* with a lipid phosphatase-deficient mutation abolished the survival advantage of *daf-18(ok480)* males (Fig. 4F), while the males of the rescue strain containing *daf-18* with a protein phosphatase-deficient mutation still possessed a survival advantage (Fig. 4G). Additionally, CRISPR/Cas9-mutated worms further support our findings, revealing that only the absence of the protein phosphatase in *daf-18(D137A)* males confers a significant survival advantage

(Fig. 4H,I). In summary, our results indicate that the survival advantage observed in *daf-18*-deficient males compared to hermaphrodites is a result of the loss of DAF-18 protein phosphatase activity.

## *unc-23* expression in *daf-18* males induces high UPS activity to eliminate proteotoxicity

Our study reveals that *daf-18(ok480)* males up-regulate *unc-23*/NEF to select and ubiquitinate unfolded proteins. The expression of *unc-23* in hermaphrodites and males was further confirmed in *daf-18(D137A)* mutants (Fig. 5A). Ubiquitination marks proteins for recognition by the proteasome, and the UPS primarily relies on substrate ubiquitination (Cundiff et al, 2019; Damgaard, 2021; Park et al, 2020). K48 branched ubiquitin chains can enhance protein degradation (Meyer and Rape, 2014), and a recent study also demonstrated that a global loss of ubiquitination across tissues impairs targeted proteasomal degradation (Koyuncu et al, 2021). To ascertain the enhanced proteasomal protein degradation in *daf-18(ok480)* males, we conducted experiments inhibiting proteasome activity with the proteasome inhibitor MG132 (Nyamsuren et al, 2007; Pataskar et al, 2022), and assessed UPS activity by monitoring the degradation of specific fluorogenic peptide substrates (ZGly-Gly-Leu-AMC) (Kisselev and Goldberg, 2005; Vilchez et al, 2012a; Vilchez et al, 2012b). Our results indicated that *daf-18 (D137A)* or *unc-23* RNAi-treated hermaphrodites exhibited low proteasome activity (Fig. 5B). In contrast, *daf-18(D137A)* males exhibited high proteasome activity, which could be reduced by *unc-23* RNAi or proteasome inhibitor MG132 treatment (Fig. 5C). These findings further substantiate the connection between the male survival advantage upon *daf-18* loss and UPS activity.

The unfolded proteins degraded by UPS activity may play a pivotal role in controlling sex-biased survival upon *daf-18* loss. As we previously demonstrated, protein aggregation in

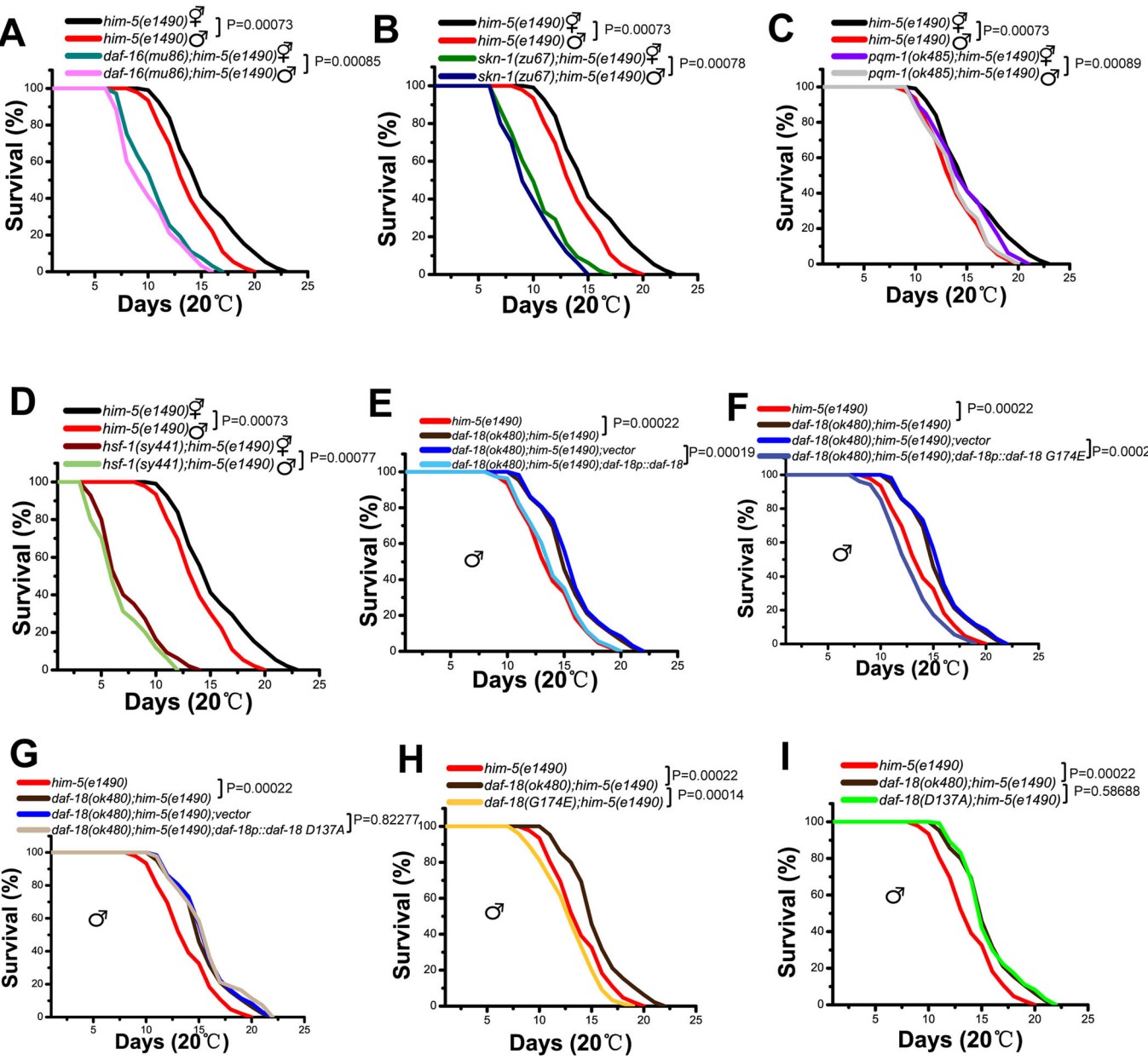

**Figure 4. DAF-18 protein phosphatase activity is critical for male survival bias.**

Comparison of the survival rates between adult hermaphrodites and males of *daf-16(mu86)*, sample size (*n*) = 92(**A**), *skn-1(zu67)*, sample size (*n*) = 91 (**B**), *pmq-1(ok485)*, sample size (*n*) = 60 (**C**), and *hsf-1(sy441)*, sample size (*n*) = 94 (**D**). (**E**) Genomic *daf-18* abolished the survival advantage of adult *daf-18* males. Sample size (*n*) = 77. (**F**) Genomic *daf-18* with the G174E mutation failed to maintain the survival advantage of adult *daf-18* males. (G174E mutation: DAF-18 lipid phosphatase-deficient mutation). Sample size (*n*) = 68. (**G**) Genomic *daf-18* with the D137A mutation retained the survival advantage of adult *daf-18* males. (D137A mutation: DAF-18 protein phosphatase-deficient mutation). Sample size (*n*) = 69. (**H**) *daf-18(G174E)* mutant males exhibited no adult survival advantage. Sample size (*n*) = 85. (**I**) Adult *daf-18(D137A)* mutant males showed a survival advantage. Sample size (*n*) = 69. Each lifespan experiment set was independently repeated three times. Mean survival rates were calculated using the Kaplan–Meier method, and *P* values were determined using the log-rank test (*P* values; **A**, *him-5(e1490)* hermaphrodites vs males: 0.00073; *daf-16(mu86);him-5(e1490)* hermaphrodites vs males:0.00085; **B**, *skn-1(zu67);him-5(e1490)* hermaphrodites vs males: 0.00078; **C**, *pmq-1(ok485);him-5(e1490)* hermaphrodites vs males: 0.00089; **D**, *hsf-1(sy441);him-5(e1490)* hermaphrodites vs males: 0.00077; **E**, *him-5(e1490)* males vs *daf-18(ok480);him-5(e1490)* males: 0.00022; *daf-18(ok480);him-5(e1490);vector* males vs *daf-18(ok480);him-5(e1490);daf-18p::daf-18* males: 0.00019; **F**, *daf-18(ok480);him-5(e1490);vector* males vs *daf-18(ok480);him-5(e1490);daf-18p::daf-18G174E* males: 0.0002; **G**, *daf-18(ok480);him-5(e1490);vector* males vs *daf-18(ok480);him-5(e1490);daf-18p::daf-18D137A* males: 0.82277; **H**, *daf-18(ok480);him-5(e1490)* males vs *daf-18(G174E);him-5(e1490)* males: 0.00014; **I**, *daf-18(ok480);him-5(e1490)* males vs *daf-18(D137A);him-5(e1490)* males: 0.58688). Source data are available online for this figure.

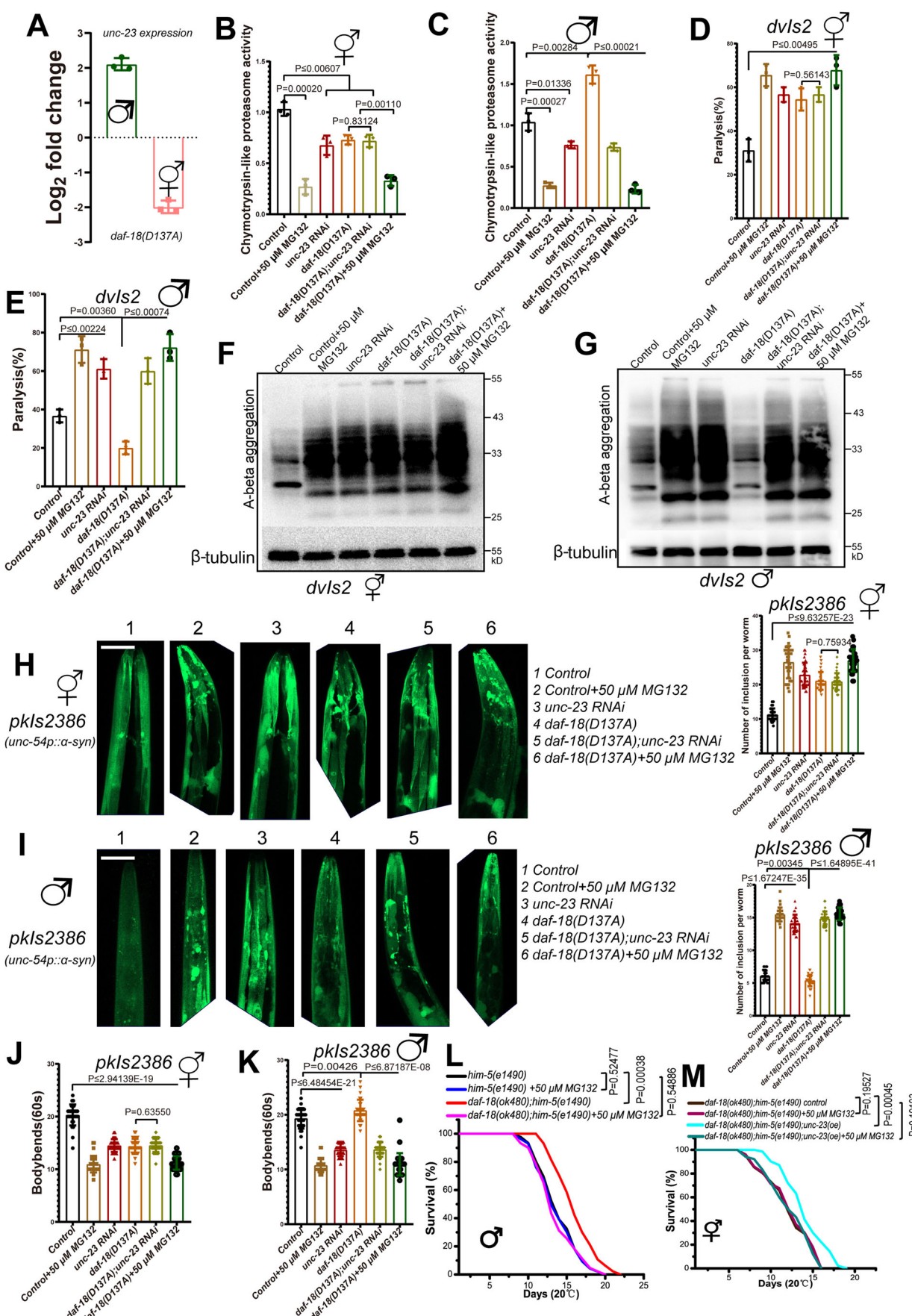

◀ **Figure 5. *daf-18 (D137A)* males exhibited higher UPS activity and lower levels of proteotoxic proteins, with a critical role played by *unc-23*.**

(A) Log2 fold changes of *unc-23* in *daf-18(D137A)* hermaphrodites and males. The data show the average of three independent repeats, and the error bars show the standard deviations. Chymotrypsin-like proteasome activity (relative slope to control) was assessed in hermaphrodites (B) and males (C), in the presence or absence of *daf-18(D137A)*, when fed with L4440 (Control) or *unc-23* RNAi. The data show the average of three independent repeats, and the error bars show the standard deviations. The *P* value was determined by using a two-tailed *t* test (*P* values; B, *control* hermaphrodites vs *control* + 50 μM MG132 hermaphrodites: 0.00020; *control* hermaphrodites vs *unc-23* RNAi hermaphrodites: 0.00607; *control* hermaphrodites vs *daf-18(D137A)* hermaphrodites: 0.00305, shown as ≤0.00607; *control* hermaphrodites vs *daf-18(D137A);unc-23* RNAi hermaphrodites: 0.00401, shown as ≤0.00607; *daf-18(D137A)* hermaphrodites vs *daf-18(D137A);unc-23* RNAi hermaphrodites: 0.83124; *daf-18(D137A)* hermaphrodites vs *daf-18(D137A)* + 50 μM MG132 hermaphrodites: 0.00110; C, *control* males vs *control* + 50 μM MG132 males: 0.00027; *control* males vs *unc-23* RNAi males: 0.01336; *control* males vs *daf-18(D137A)* males: 0.00284; *daf-18(D137A)* males vs *daf-18(D137A);unc-23* RNAi males: 0.00021; *daf-18(D137A)* males vs *daf-18(D137A)* + 50 μM MG132 males: 3.93763E-05, shown as ≤0.00021. Percentage of paralysis in day 5 CL2006, *dvIs2* [*Punc-54::human A-beta 3-42; pRF4 (rol-6(su1006))*] hermaphrodites (D) and males (E), in the presence or absence of *daf-18(D137A)*, when fed with L4440 (Control) or *unc-23* RNAi. The data show the average of three independent repeats, and the error bars show the standard deviations. The *P* value was determined by using a two-tailed *t* test (*P* values; D, *control* hermaphrodites vs *control* + 50 μM MG132 hermaphrodites: 0.00115, shown as ≤0.00495; *control* hermaphrodites vs *unc-23* RNAi hermaphrodites: 0.00189, shown as ≤0.00495; *control* hermaphrodites vs *daf-18(D137A)* hermaphrodites: 0.00495; *control* hermaphrodites vs *daf-18(D137A);unc-23* RNAi hermaphrodites: 0.00189, shown as ≤0.00495; *control* hermaphrodites vs *daf-18(D137A)* + 50 μM MG132 hermaphrodites: 0.00179, shown as ≤0.00495; *daf-18(D137A)* hermaphrodites vs *daf-18(D137A);unc-23* RNAi hermaphrodites: 0.56143; E, *control* males vs *control* + 50 μM MG132 males: 0.00149, shown as ≤0.00224; *control* males vs *unc-23* RNAi males: 0.00224; *control* males vs *daf-18(D137A)* males: 0.00360; *daf-18(D137A)* males vs *daf-18(D137A);unc-23* RNAi males: 0.00074; *daf-18(D137A)* males vs *daf-18(D137A)* + 50 μM MG132 males: 0.00030, shown as ≤0.00074). Western blots of A-beta aggregation in day 5 of adulthood *dvIs2* hermaphrodites (F) and males (G), in the presence or absence of *daf-18(D137A)*, when fed with L4440 (Control) or *unc-23* RNAi. Alpha-synuclein expression in the head region and the average number of inclusions per animal between the tip of the nose and pharyngeal bulb of hermaphrodites (H) or males (I) at day 5 of adulthood, in the presence or absence of *daf-18(D137A)*, when fed with L4440 (Control) or *unc-23* RNAi. Scale bar: 50 μm. The data shows the values of all samples from three replicates, with error bars representing the averages and standard deviations. The *P* value was determined by using a two-tailed *t* test (*P* values; H, *control* hermaphrodites vs *control* + 50 μM MG132 hermaphrodites: 1.01734E-23, shown as ≤9.63257E-23; *control* hermaphrodites vs *unc-23* RNAi hermaphrodites: 9.63257E-23; *control* hermaphrodites vs *daf-18(D137A)* hermaphrodites: 2.99212E-24, shown as ≤9.63257E-23; *control* hermaphrodites vs *daf-18(D137A);unc-23* RNAi hermaphrodites: 1.36426E-24, shown as ≤9.63257E-23; *control* hermaphrodites vs *daf-18(D137A)* + 50 μM MG132 hermaphrodites: 4.61549E-31, shown as ≤9.63257E-23; *daf-18(D137A)* hermaphrodites vs *daf-18(D137A);unc-23* RNAi hermaphrodites: 0.75934; I, *control* males vs *control* + 50 μM MG132 males: 6.84324E-43, shown as ≤1.67247E-35; *control* males vs *unc-23* RNAi males: 1.67247E-35; *control* males vs *daf-18(D137A)* males: 0.00345; *daf-18(D137A)* males vs *daf-18(D137A);unc-23* RNAi males: 1.64895E-41; *daf-18(D137A)* males vs *daf-18(D137A)* + 50 μM MG132 males: 1.37243E-45, shown as ≤1.64895E-41). The mobility of hermaphrodites (J) and males (K), in the presence or absence of *daf-18(D137A)*, when fed with L4440 (Control) or *unc-23* RNAi, was assessed using thrashing experiments. Animals that were alive but unable to swim were scored as paralyzed. Motility assays were carried out by counting body bends over a 30-second interval in M9 at day 5. The data shows the values of all samples from three replicates, with error bars representing the averages and standard deviations. The *P* value was determined by using a two-tailed *t* test (*P* values; J, *control* hermaphrodites vs *control* + 50 μM MG132 hermaphrodites: 5.93628E-28, shown as ≤2.94139E-19; *control* hermaphrodites vs *unc-23* RNAi hermaphrodites: 1.01817E-19, shown as ≤2.94139E-19; *control* hermaphrodites vs *daf-18(D137A)* hermaphrodites: 2.94139E-19; *control* hermaphrodites vs *daf-18(D137A);unc-23* RNAi hermaphrodites: 2.27463E-19, shown as ≤2.94139E-19; *control* hermaphrodites vs *daf-18(D137A)* + 50 μM MG132 hermaphrodites: 1.78265E-29, shown as ≤2.94139E-19; *daf-18(D137A)* hermaphrodites vs *daf-18(D137A);unc-23* RNAi hermaphrodites: 0.63550; K, *control* males vs *control* + 50 μM MG132 males: 1.08379E-28, shown as ≤6.48454E-21; *control* males vs *unc-23* RNAi males: 6.48454E-21; *control* males vs *daf-18(D137A)* males: 0.00426; *daf-18(D137A)* males vs *daf-18(D137A);unc-23* RNAi males: 1.19793E-23, shown as ≤6.87187E-08; *daf-18(D137A)* males vs *daf-18(D137A)* + 50 μM MG132 males: 6.87187E-08). (H–K). Worms carried the *him-5(e1490)* mutation to ensure sufficient males were produced. Sample size (*n*) = 30. (L) Treating *daf-18(ok480)* males with proteasome function inhibitor MG132 led to a reduction in adult lifespan. Sample size (*n*) = 83. (M) The improved adult lifespan of *daf-18(ok480)* hermaphrodites achieved through overexpression of *unc-23* was reduced when treated with proteasome function inhibitor MG132. Sample size (*n*) = 105. Each lifespan experiment set was independently repeated three times. Mean survival rates were calculated using the Kaplan–Meier method, and *P* values were determined using the log-rank test (*P* values; L, *him-5(e1490)* males vs *him-5(e1490)*+50 μM MG132 males: 0.52477; *him-5(e1490)* males vs *daf-18(ok480);him-5(e1490)* males:0.00038; *him-5(e1490)*+50 μM MG132 males vs *daf-18(ok480);him-5(e1490)*+50 μM MG132 males:0.54886; M, *daf-18(ok480);him-5(e1490)* hermaphrodites vs *daf-18(ok480);him-5(e1490)*+50 μM MG132 hermaphrodites: 0.19527; *daf-18(ok480);him-5(e1490)* hermaphrodites vs *daf-18(ok480);him-5(e1490);unc-23(oe)* hermaphrodites: 0.00045; *daf-18(ok480);him-5(e1490);unc-23(oe)* +50 μM MG132 hermaphrodites vs *daf-18(ok480);him-5(e1490)* hermaphrodites: 0.19193). Source data are available online for this figure.

hermaphrodites is higher than in males when *daf-18* is absent. To further confirm this, we conducted proteotoxicity analyses in both males and hermaphrodites using the CL2006 (*dvIs2*) strain (associated with Alzheimer's disease-related proteotoxicity of A-beta peptide) (Volovik et al, 2014). Our results revealed that the percentage of paralysis and A-beta aggregation in *dvIs2* hermaphrodites increased due to the *daf-18(D137A)* mutation or *unc-23* RNAi treatment (Fig. 5D,F). However, *daf-18(D137A)* males exhibited lower paralysis rates and reduced A-beta aggregation, which could be increased by *unc-23* RNAi or proteasome inhibitor MG132 treatment (Fig. 5E,G). We also used the NL5901 (*pkIs2386*) strain (expressing the pathological hallmark of Parkinson's disease, Alpha-synuclein, in body wall muscles) to assess proteotoxicity in both males and hermaphrodites (Kim et al, 2008). The results indicated that aggregated Alpha-synuclein in hermaphrodites increased due to the *daf-18(D137A)* mutation or *unc-23* RNAi (Fig. 5H). In contrast, *daf-18(D137A)* males had lower levels of aggregated Alpha-synuclein, which could be increased by *unc-23* RNAi or MG132 (Fig. 5I). Furthermore, motility tests showed that hermaphrodites experienced decreased motility due to the *daf-*

*18(D137A)* mutation or *unc-23* RNAi (Fig. 5J). Conversely, *daf-18(D137A)* males exhibited higher motility, but this could be decreased by *unc-23* RNAi or MG132 treatment (Fig. 5K). To further confirm the role of proteasome activity, MG132 was used to treat *daf-18(ok480)* males and *unc-23* overexpression hermaphrodites, which resulted in the abolishment of the survival advantage in males (Fig. 5L) and *daf-18(ok480);unc-23(oe)* hermaphrodites (Fig. 5M). These results provide further evidence that the survival advantage of these worms is related to the proteasome protein degradation function.

In summary, all of these in vitro and in vivo studies support the idea that *daf-18* males maintain lower proteotoxicity by regulating *unc-23*, thereby securing a survival advantage compared to hermaphrodites.

## DAF-18 dephosphorylates C18E9.2 to keep protein hemostasis

To delve deeper into how DAF-18 protein phosphatase activity governs protein homeostasis to confer a survival advantage in

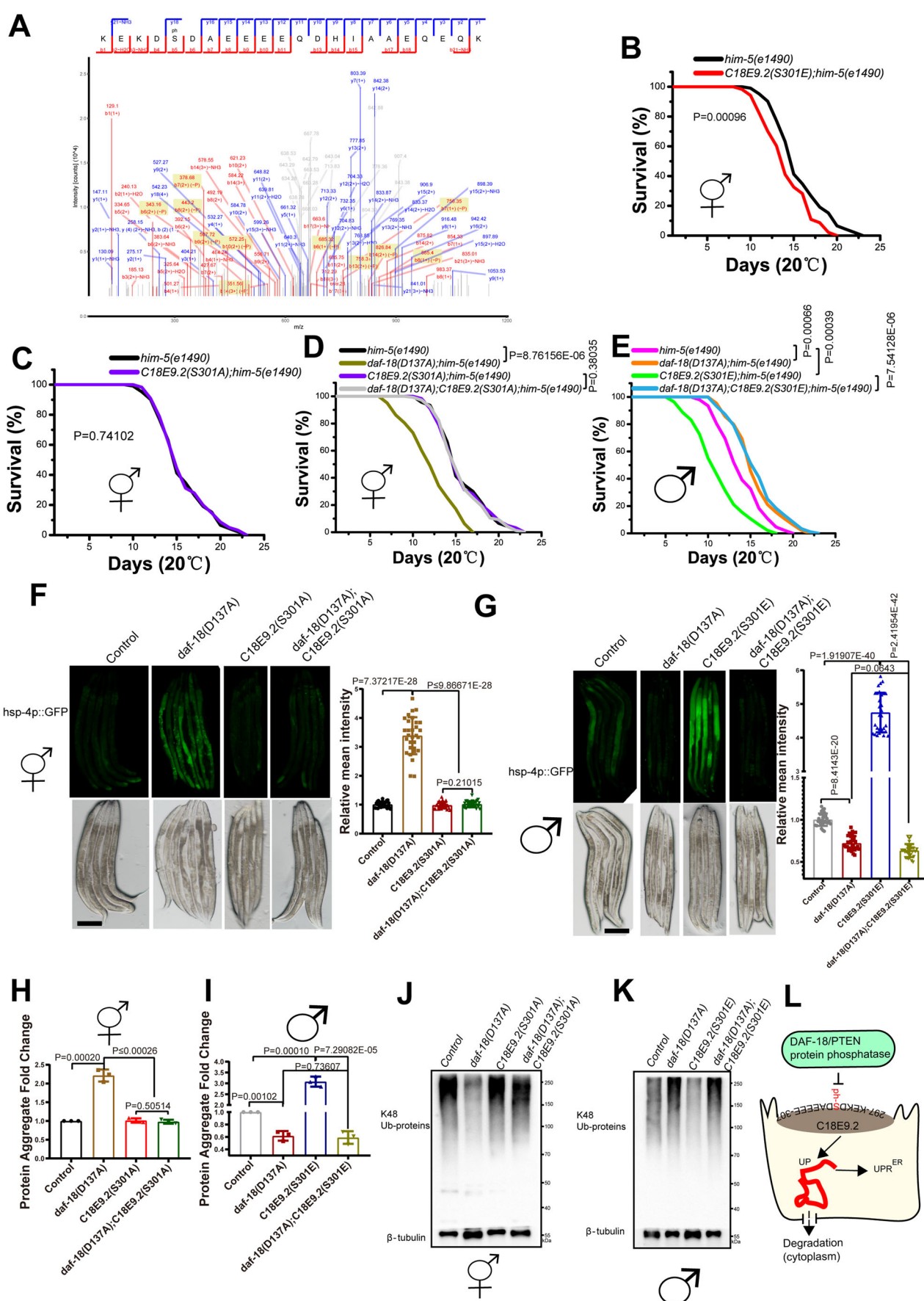

**Figure 6. The high protein aggregation level in *daf-18(D137A)* hermaphrodites, caused by phosphorylated C18E9.2, was reduced in *daf-18(D137A)* males.**

(A) Protein spectrum results for serine 301 of C18E9.2 in *daf-18(D137A)* worms compared to N2 worms. (B) Adult lifespan of *C18E9.2(S301E)* mutants. (S301E: Serine 301 of C18E9.2 changed to glutamic acid to mimic phosphorylation.). Sample size ($n$) = 84. (C) Adult lifespan of *C18E9.2(S301A)* mutants. (S301A: Serine 301 of C18E9.2 changed to alanine to prevent potential phosphorylation.). Sample size ($n$) = 75. (D) Adult lifespan of *daf-18(D137A)* hermaphrodites with (sample size $n$ = 66) and without (sample size $n$ = 95) the *C18E9.2(S301A)* mutation. (E) Adult lifespan of *daf-18(D137A)* males with (sample size $n$ = 104) and without (sample size $n$ = 69) the *C18E9.2(S301E)* mutation. Each lifespan experiment set was independently repeated three times. Mean survival rates were calculated using the Kaplan–Meier method, and $P$ values were determined using the log-rank test ($P$ values; **B**, *him-5(e1490)* hermaphrodites vs *C18E9.2(S301E);him-5(e1490)* hermaphrodites: 0.00096; **C**, *him-5(e1490)* hermaphrodites vs *C18E9.2(S301A);him-5(e1490)* hermaphrodites: 0.74102; **D**, *him-5(e1490)* hermaphrodites vs *daf-18(D137A);him-5(e1490)* hermaphrodites: 8.76156E-06; *C18E9.2(S301A);him-5(e1490)* hermaphrodites vs *daf-18(D137A);C18E9.2(S301A);him-5(e1490)* hermaphrodites: 0.38035; **E**, *him-5(e1490)* males vs *daf-18(D137A);him-5(e1490)* males: 0.00066; *him-5(e1490)* males vs *C18E9.2(S301E);him-5(e1490)* males: 0.00039; *C18E9.2(S301E);him-5(e1490)* males vs *daf-18(D137A);C18E9.2(S301E);him-5(e1490)*: 7.54128E-06). UPR$^{ER}$ in *daf-18(D137A)* hermaphrodites in the absence or presence of the *C18E9.2(S301A)* (**F**), and males in the absence or presence of the *C18E9.2(S301E)* mutation (**G**). Each experimental set had three independent repeats, and the sample size ($n$) = 30. The data shows the values of all samples from three replicates, with error bars representing the averages and standard deviations. The $P$ value was determined by using a two-tailed $t$ test ($P$ values; **F**, *control* hermaphrodites vs *daf-18(D137A)* hermaphrodites: 7.37217E-28; *daf-18(D137A)* hermaphrodites vs *C18E9.2(S301A)* hermaphrodites: 4.9185E-28, shown as ≤9.86671E-28; *daf-18(D137A)* hermaphrodites vs *daf-18(D137A);C18E9.2(S301A)* hermaphrodites: 9.86671E-28; *C18E9.2(S301A)* hermaphrodites vs *daf-18(D137A);C18E9.2(S301A)* hermaphrodites: 0.21015; **G**, *control* males vs *daf-18(D137A)* males: 8.4143E-20; *control* males vs *C18E9.2(S301E)* males: 1.91907E-40; *daf-18(D137A)* males vs *daf-18(D137A);C18E9.2(S301E)* males: 0.0643; *C18E9.2(S301E)* males vs *daf-18(D137A);C18E9.2(S301E)* males: 2.41954E-42). Scale bar: 200 μm. Fold changes of total protein aggregates in *daf-18 (D137A)* hermaphrodites in the absence or presence of the *C18E9.2(S301A)* mutation (**H**), and males in the absence or presence of the *C18E9.2(S301E)* mutation (**I**). The data show the average of three independent repeats, and the error bars show the standard deviations. The $P$ value was determined by using a two-tailed $t$ test ($P$ values; **H**, *control* hermaphrodites vs *daf-18(D137A)* hermaphrodites: 0.00020; *daf-18(D137A)* hermaphrodites vs *C18E9.2(S301A)* hermaphrodites: 0.00026; *daf-18 (D137A)* hermaphrodites vs *daf-18(D137A);C18E9.2(S301A)* hermaphrodites: 0.00023, shown as ≤0.00026; *C18E9.2(S301A)* hermaphrodites vs *daf-18(D137A);C18E9.2(S301A)* hermaphrodites: 0.50514; **I**, *control* males vs *daf-18(D137A)* males: 0.00102; *control* males vs *C18E9.2(S301E)* males: 0.00010; *daf-18(D137A)* males vs *daf-18(D137A);C18E9.2(S301E)* males: 0.73607; *C18E9.2(S301E)* males vs *daf-18(D137A);C18E9.2(S301E)* males: 7.29082E-05). (**J**) K48-linked protein ubiquitination of *daf-18(D137A)* and *C18E9.2(S301A)* hermaphrodites. (**K**) K48-linked protein ubiquitination of *daf-18(D137A)* and *C18E9.2(S301E)* males. (**L**) The proposed model suggests that DAF-18 dephosphorylates C18E9.2 to inhibit the production of unfolded proteins in ER. The unfolded and misfolded proteins are secreted into cytoplasm to be degraded. UP unfolded proteins. UPR$^{ER}$: ER unfolded protein response. (**F–K**) Control: *him-5(e1490)* worms. Other mutants also possess this *him-5(e1490)* mutation. Each immunoblot is representative of three independent experiments. Source data are available online for this figure.

males, we conducted a comparative analysis of target proteins dephosphorylated by DAF-18 in N2 and *daf-18(D137A)* (protein phosphatase deficient) worms. Our findings revealed significant phosphorylation at serine 301 of C18E9.2, a protein localized on the ER membrane, in *daf-18(D137A)* worms (Fig. 6A). Given that the loss of DAF-18 protein phosphatase resulted in elevated UPR$^{ER}$ but not in the mitochondria or cytosol (Appendix Figure S6), we hypothesized that DAF-18 directly or indirectly dephosphorylates C18E9.2 to maintain ER functional integrity. Indeed, hermaphrodites with the *C18E9.2 (S301E)* mutation, mimicking constitutive phosphorylation of the serine residue, exhibited a shortened adult lifespan (Fig. 6B), whereas *C18E9.2(S301A)* hermaphrodites, preventing phosphorylation of the serine residue, showed no significant changes compared to control worms (Fig. 6C), and overexpressing *unc-23* in *C18E9.2(S301E)* can extend the lifespan of *C18E9.2(S301E)* hermaphrodites (Fig. EV5). Furthermore, we found that the *C18E9.2(S301A)* mutation could rescue the shortened adult lifespan of *daf-18(D137A)* hermaphrodites (Fig. 6D), These results affirmed that DAF-18 dephosphorylates C18E9.2 to ensure worm survival. Intriguingly, the reduced adult lifespan observed in *C18E9.2(S301E)* males was significantly extended by the *daf-18(D137A)* mutation (Fig. 6E), aligning with our earlier finding that *daf-18(ok480)* males exhibit a survival advantage. Moreover, we observed a high level of UPR$^{ER}$ (Fig. 6F) and protein aggregation (Fig. 6H) in *daf-18(D137A)* hermaphrodites, which was significantly diminished by the *C18E9.2(S301A)* mutation. Conversely, the elevated levels of UPR$^{ER}$ (Fig. 6G) and protein aggregation (Fig. 6I) in *C18E9.2(S301E)* males were reduced by the *daf-18(D137A)* mutation. We demonstrated that *daf-18*-deficient males exhibit a high level of K48 ubiquitination to process unfolded proteins and reduce protein aggregation. Next, we assessed whether the K48 ubiquitination level is affected by *daf-18(D137A)* and *C18E9.2* in hermaphrodites and males. The results

are consistent with our findings in *daf-18(ok480)* worms, where *daf-18(D137A)* hermaphrodites displayed lower K48 ubiquitination levels (Fig. 6J). *daf-18(D137A);C18E9.2(S301A)* mutants also exhibited lower K48 ubiquitination levels compared to *C18E9.2(S301A)* mutants (Fig. 6J). This confirms that *daf-18*-deficient hermaphrodites have a reduced ability to digest aggregated protein. *C18E9.2(S301A)* mutants showed no significant change compared to control (Fig. 6K), confirming that DAF-18 dephosphorylates C18E9.2 to maintain normal unfolded protein levels and subsequently control protein aggregation levels. Moreover, *C18E9.2(S301E)* males exhibited lower K48 ubiquitination levels, while, akin to *daf-18(ok480)* males, *daf-18(D137A)* males maintained high levels of K48 ubiquitination, regardless of the presence or absence of *C18E9.2(S301E)* (Fig. 6K). These cumulative findings suggest that the phosphorylation of C18E9.2, caused by the loss of DAF-18 protein phosphatase, can trigger a high level of unfolded proteins, UPR$^{ER}$ (Fig. 6L) and global protein aggregation. Nevertheless, *daf-18*-deficient males can up-regulate the ubiquitination system to enhance protein degradation, thereby securing a survival advantage.

## Male sex determination is evoked by *daf-18* loss in male worms to maintain protein homeostasis

Next, we investigated why *daf-18(ok480)* males exhibited upregulated expression of *unc-23*, which is opposite to the expression pattern in hermaphrodites. As survival is related to sex differences, and as we found that the differentially expressed genes in *daf-18(ok480)* males were enriched with the gametes generation and sexual reproduction Gene Ontology (GO) terms (Appendix Fig. S7), we examined the sex determination pathway genes (Fig. 7A) in the RNA-seq data and found that sex determination genes were significantly changed in *daf-18(ok480)* males (Fig. 7B). The downregulated expression of *tra-3* and

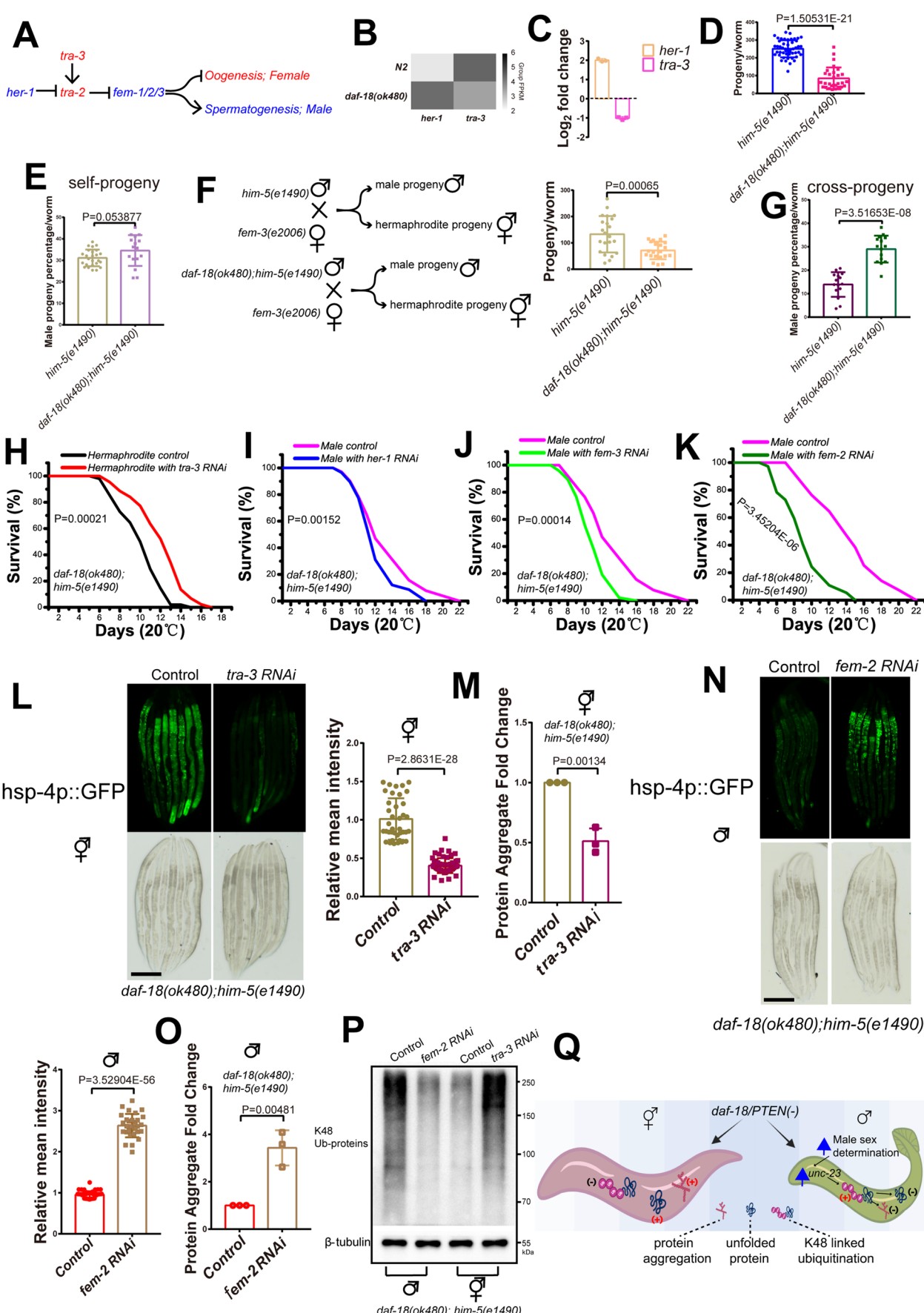

◀ **Figure 7. *daf-18* loss evokes male sex determination to maintain protein homeostasis in males.**

(A) Sex determination pathway affecting the germline and somatic cells. Blue color: male sex determination signal. Red color: hermaphroditic sex determination signal. (B) The average expression of the male sex determination genes *her-1* and *tra-3* was altered in *daf-18(ok480)* males according to RNA-seq data. Group FPKM: average of the log-scaled FPKM: log2(FPKM + 1) of the gene in groups. (C) Real-time PCR confirmed the expression changes of *her-1* and *tra-3* in males. All these genes were significantly up- or downregulated by at least 2-fold. Log2FC: log2(fold change) ( >1 or <−1). The data show the average of three independent repeats, and the error bars show the standard deviations. (D) Total progeny numbers of self-fertilized hermaphrodites. Sample size (*n*) = 49 (*him-5*), 29 (*daf-18*). The data shows the values of all samples from three replicates, with error bars representing the averages and standard deviations. The *P* value was determined by using a two-tailed *t* test (*P* value: 1.50531E-21). (E) Percentage of males in total progeny of self-fertilized hermaphrodites. Sample size (*n*) = 25 (*him-5*), 16 (*daf-18*). The data shows the values of all samples from three replicates, with error bars representing the averages and standard deviations. The *P* value was determined by using a two-tailed *t* test (*P* value: 0.053877). (F) *him-5(e1490)* and *daf-18(ok480);him-5(e1490)* males were mated with *fem-3(e2006)* (which can only provide oocytes). The total progeny numbers of the cross experiments are shown. Sample size (n)=22 (*him-5*), 21 (*daf-18*). The data shows the values of all samples from three replicates, with error bars representing the averages and standard deviations. The *P* value was determined by using a two-tailed *t* test (*P* value: 0.00065). (G) Percentage of males in the total progeny of crosses of *daf-18(ok480);him-5(e1490)* males with *fem-3(e2006)*. Sample size (n)=15 (*him-5*), 15 (*daf-18*). The data shows the values of all samples from three replicates, with error bars representing the averages and standard deviations. The *P* value was determined by using a two-tailed *t* test (*P* value: 3.51653E-08). (H) Knocking down *tra-3* extended the lifespan of *daf-18(ok480)* hermaphrodites. Sample size (*n*) = 98 (*control*), 90 (*tra-3*). (I) Knocking down *her-1*, (J) *fem-3* or (K) *fem-2* reduced the lifespan of *daf-18(ok480)* males. Sample size (*n*) = 102 (*control*), 108 (*her-1*), 94 (*fem-3*) and 78 (*fem-2*). Each lifespan experiment was repeated three times. The mean survival rates were calculated using the Kaplan–Meier method, and *P* values were determined by using the log rank test (*P* values; H, *control* hermaphrodites vs *tra-3* RNAi hermaphrodites: 0.00021; I, *control* males vs *her-1* RNAi males: 0.00152; J, *control* males vs *fem-3* RNAi males: 0.00014; K, *control* males vs *fem-2* RNAi males:3.45204E-06.) (L) Unfolded protein in *daf-18(ok480)* hermaphrodites when *tra-3* was knocked down. Sample size (*n*) = 38 (*control*), 59 (*tra-3*). The data shows the values of all samples from three replicates, with error bars representing the averages and standard deviations. The *P* value was determined by using a two-tailed *t* test (*P* value: 2.8631E-28). Scale bar: 200 μm. (M) The average fold changes of protein aggregation in hermaphrodites in three independent experiments. The data show the average of three independent repeats, and the error bars show the standard deviations. The *P* value was determined by using a two-tailed *t* test (*P* value: 0.00134). (N) Unfolded protein in *daf-18(ok480)* males when *fem-2* was knocked down. Sample size (*n*) = 54 (*control*), 30 (*fem-2*). The data shows the values of all samples from three replicates, with error bars representing the averages and standard deviations. The *P* value was determined by using a two-tailed *t* test (*P* value: 3.52904E-56). Scale bar: 200 μm. (O) Fold changes in protein aggregation in males in three independent experiments. The data show the average of three independent repeats, and the error bars show the standard deviations. The *P* value was determined by using a two-tailed *t* test (*P* value: 0.00481). (P) K48-linked protein ubiquitination in males and hermaphrodites when sex determination signaling was changed. The immunoblot is representative of three independent experiments. Controls in (H–P): RNAi control clones containing the empty vector L4440. (Q) Model of the mechanism by which *daf-18*-deficient males evoke male sex determination to maintain survival according to our results. When *daf-18* is lost, males can evoke the male sex determination pathway to up-regulate the *unc-23*/NEF-related protein ubiquitination function in order to maintain protein homeostasis. Blue arrows: enhancing signaling or up-regulation. ( + ): more, (−): less. Source data are available online for this figure.

upregulated expression of *her-1* in *daf-18(ok480)* males were confirmed by using real-time PCR (Fig. 7C). Changes in the expression of sex determination genes can increase sperm generation and male progeny in males (Ellis and Schedl, 2007; Zarkower, 2006). TRA-3 likely promotes female development in XX hermaphrodites by cleaving the membrane-associated TRA-2A protein to generate a TRA-2 peptide that likely has feminizing activity (Barnes and Hodgkin, 1996b). Therefore, we tested the progeny of *daf-18(ok480)* hermaphrodites and males. We found that the total progeny of *daf-18(ok480)* worms was less than those of the controls (Fig. 7D,F), and the percentages of male progeny were not significantly changed by *daf-18* loss (Fig. 7E). However, the male progeny generated from *daf-18(ok480)* male parents mating with *fem-3(e2006)* hermaphrodites (only providing oocytes) were higher than those generated from wild-type mating (Fig. 7G). These results suggest that loss of *daf-18* may evoke male sex determination signaling. To test this speculation, we examined the effects of *her-1*, *fem-2/3* (which were also confirmed to be upregulated in *daf-18(ok480)* males, Appendix Fig. S8) and *tra-3* on the sex-specific differences in the survival of *daf-18(ok480)* mutants. We found that knocking down *tra-3* to activate the male sex determination signal in hermaphrodites reversed the shortening of adult lifespan (Fig. 7H). In contrast, knocking down *her-1* (Fig. 7I) or *fem-2/3* (Fig. 7J,K) to weaken the male sex determination signal reduced the adult lifespan of *daf-18(ok480)* males, with *fem-2* RNAi yielding the best output. Furthermore, these adult lifespan data affected by knocking down *tra-3*, *her-1* and *fem-2/3* were confirmed by using mutants without *him-5(e1490)* (Appendix Fig. S9 and Appendix Table S1) and individually culturing experiments (Appendix Fig. S10 and Appendix Table S3). Next, we tested whether proteostasis is regulated by these sex determination genes in *daf-18(ok480)* mutants.

We found that the UPR^ER marker *hsp-4p*::GFP (Fig. 7L) and protein aggregation levels (Fig. 7M) were all reduced by knocking down *tra-3* in *daf-18(ok480)* hermaphrodites; in contrast, they were increased by knocking down *fem-2* in *daf-18(ok480)* males (Fig. 7N,O). We also tested protein degradation by ubiquitination, and the results showed that the K48-linked protein ubiquitination levels were increased by knocking down *tra-3* in *daf-18(ok480)* hermaphrodites (Fig. 7P); in contrast, they were reduced by knocking down *fem-2* in *daf-18*-deficient males (Fig. 7P). All these results show that males can counteract *daf-18* loss by evoking the male sex determination pathway (Fig. 7Q).

## The expression of *unc-23* is regulated by the global sex determination pathway

According to all these results, we speculate that the *daf-18(ok480)* males may up-regulate the expression of *unc-23* through the male sex determination pathway to maintain survival. Next, we tested whether the expression of *unc-23* could be regulated by *tra-3* and *fem-2* in *daf-18(ok480)* hermaphrodites and males, respectively. We found that *unc-23* expression could be downregulated by knocking down *fem-2* or *her-1* in *daf-18(ok480)* males and upregulated by knocking down *tra-2/3* in *daf-18(ok480)* hermaphrodites (Figs. 8A and EV6A), which suggests that *unc-23* may be a target gene of the male sex determination pathway. To confirm this speculation, we knocked down *tra-2/3* in *daf-18(ok480);unc-23(e25)* double mutants and overexpressed *unc-23* in *daf-18(ok480);fem-2(b245)* or *daf-18(ok480);her-1(n695)* double mutants. We found that *tra-2/3* RNAi failed to rescue the shortened adult lifespan (Figs. 8B and EV6B) or reduce the UPR^ER (Fig. 8C) and protein aggregation levels

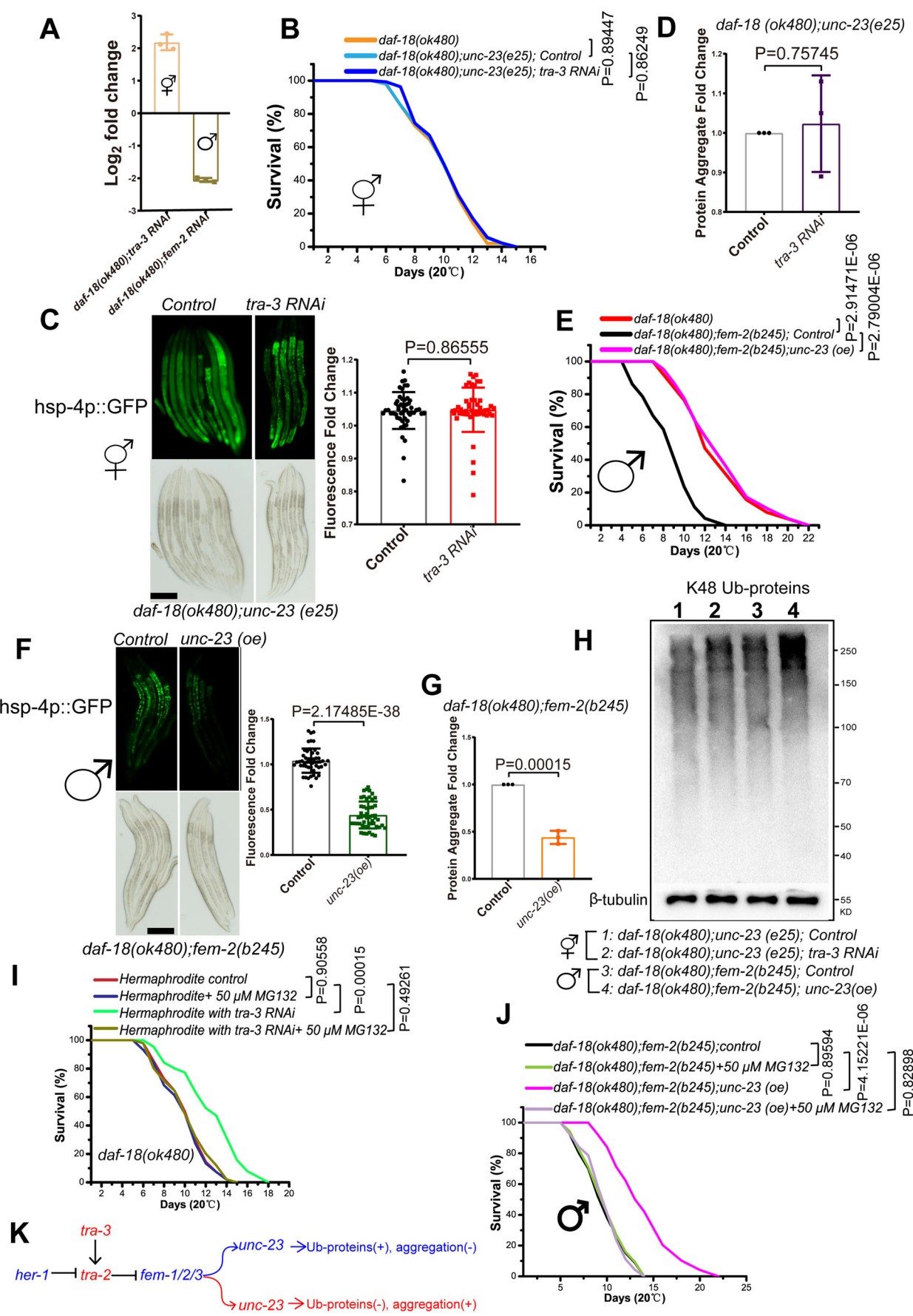

◀  **Figure 8.  *unc-23* may be the target gene of male sex determination pathway.**

(A) Real-time PCR tested the expression changes of *unc-23*. All these genes were significantly up- or downregulated by at least twofold. Log2FC: log2 (fold change) ( >1 or <−1). The data show the average of three independent repeats, and the error bars show the standard deviations. (B) Knocking down *tra-3* failed to extend the adult lifespan *daf-18(ok480);unc-23(e25)* hermaphrodites. The *tra-3* knocking down efficiency in *daf-18(ok480);unc-23(e25)* hermaphrodites was confirmed. Sample size (*n*) = 111 (*control*), 95 (*tra-3*). Each lifespan experiment set was independently repeated three times. *P* values were determined using the log-rank test (*P* values; *daf-18(ok480)* hermaphrodites vs *daf-18 (ok480);unc-23(e25)* hermaphrodites: 0.89447; *daf-18(ok480);unc-23(e25)* hermaphrodites vs *daf-18(ok480);unc-23(e25);tra-3 RNAi* hermaphrodites: 0.86249). (C) The unfolded protein marker *hsp-4p::GFP* in *daf-18(ok480);unc-23(e25)* hermaphrodites. The data shows the values of all samples from three replicates, with error bars representing the averages and standard deviations. The *P* value was determined by using a two-tailed *t* test (*P* value: 0.86555). Scale bar: 200 μm. Sample size (*n*) = 50. (D) The average fold changes of protein aggregation in *daf-18(ok480);unc-23(e25)* of three independent experiments. The data show the average of three independent repeats, and the error bars show the standard deviations. The *P* value was determined by using a two-tailed *t* test (*P* value: 0.75745). (B–D) Control: Feeding RNAi control clones with empty vector L4440. (E) As *unc-23* was downregulated when disrupting *fem-2*, overexpressed *unc-23* in *daf-18(ok480);fem-2(b245)* extended the lifespan. Sample size (*n*) = 67 (*control*), 86 (*unc-23 oe*). Each lifespan experiment set was independently repeated three times. *P* values were determined using the log-rank test (*P* values; *daf-18(ok480)* males vs *daf-18(ok480);fem-2(b245)* males: 2.91471E-06; *daf-18(ok480);fem-2(b245)* males vs *daf-18 (ok480);fem-2(b245);unc-23(oe)* males: 2.79004e-06). (F) The unfolded protein marker *hsp-4p::GFP* in *daf-18(ok480); fem-2 (b245)* males. Scale bar: 200 μm. Sample size (*n*) = 50. The data shows the values of all samples from three replicates, with error bars representing the averages and standard deviations. The *P* value was determined by using a two-tailed *t* test (*P* value: 2.17485E-38). (G) The fold changes of protein aggregation in *daf-18(ok480); fem-2(b245)* males of three independent experiments. The data show the average of three independent repeats, and the error bars show the standard deviations. The *P* value was determined by using a two-tailed *t* test (*P* value: 0.00015). (E–G) Control: the transgenic injection strains with empty expression vector L2528. (H) The k48 linked protein ubiquitination. Each immunoblot shows representative of three times independent experiments. (I) The improved adult lifespan of *daf-18(ok480)* hermaphrodites achieved through knocking down *tra-3* to enhance male sex determination signaling was also reduced when treated with proteasome function inhibitor MG132. Sample size (*n*) = 96 (*control*), 93 (*control + MG132*), 73 (*tra-3*), 94 (*tra-3 + MG132*). (J) The improved adult lifespan of *daf-18(ok480);fem-2(b245)* males achieved through overexpression of *unc-23* was also reduced when treated with proteasome function inhibitor MG132. Sample size (*n*) = 87 (*control*), 105 (*control + MG132*), 108 (*unc-23 oe*), 96 (*unc-23 oe + MG132*). Each lifespan experiment set was repeated three times. The mean survival rates were calculated using the Kaplan–Meier method, and *P* values were carried out by using log rank test (*P* values; I, *control* hermaphrodites vs *control + 50 μM* MG132 hermaphrodites: 0.90558; *control* hermaphrodites vs *tra-3 RNAi* hermaphrodites: 0.00015; *control* hermaphrodites+*50 μM* MG132 vs *tra-3 RNAi + 50 μM* MG132 hermaphrodites: 0.49261; J, *daf-18(ok480);fem-2(b245)* males vs *daf-18(ok480);fem-2(b245)+50 μM* MG132 males: 0.89594; *daf-18(ok480);fem-2(b245)* males vs *daf-18(ok480);fem-2(b245);unc-23(oe)* males: 4.15221E-06; *daf-18(ok480);fem-2(b245)* males+*50 μM* MG132 vs *daf-18(ok480);fem-2(b245);unc-23(oe)+50 μM* MG132 males: 0.82898). (H–J) Control for RNAi: Feeding RNAi control clones with empty vector L4440. Control for overexpression: the transgenic injection strains with empty expression vector L2528. (K) The male sex determination signaling regulate the unfolded proteins and protein aggregation through *unc-23*. Source data are available online for this figure.

(Figs. 8D and EV6D) of *daf-18(ok480);unc-23(e25)* hermaphrodites. In contrast, overexpression of *unc-23* in *daf-18(ok480);fem-2(b245)* or *daf-18(ok480);her-1(n695)* double mutants extended the adult lifespan (Figs. 8E and EV6C) and reduced the UPR^ER (Fig. 8F) and protein aggregation levels (Figs. 8G and EV6E). We also found that K48-linked protein ubiquitination was not further changed by disrupting *tra-2/3* in *daf-18(ok480);unc-23(e25)* hermaphrodites, but overexpression of *unc-23* enhanced the K48-linked protein ubiquitination of *daf-18(ok480);fem-2(b245)* or *daf-18(ok480);her-1(n695)* male worms (Figs. 8H and EV6F). To further ascertain the enhanced proteasomal protein degradation in *daf-18(ok480)* males, we conducted experiments inhibiting proteasome activity with the proteasome inhibitor MG132, which resulted in the abolishment of the survival advantage in *daf-18(ok480)* hermaphrodites with enhanced male sex determination signaling (Fig. 8I) and *daf-18(ok480);fem-2(b245);unc-23(oe)* males (Fig. 8J). These results provide further evidence that the survival advantage of these worms is related to the proteasome protein degradation function. Taken together, our results suggest that the effects of UNC-23 on protein hemostasis may be regulated by male sex determination (Fig. 8K).

## Enhancing male sex determination reduces proteotoxicity in *daf-18* worms

Our results demonstrate that male sex determination is activated in *daf-18(ok480)* males, and this finding was further validated in *daf-18(D137A)* males (Fig. 9A). Unfolded proteins can cause protein aggregation in worms, leading to proteotoxicity that influences survival and disease (Walther et al, 2015). To further ascertain

whether male sex determination signaling regulates the degradation of unfolded and aggregated proteins, we investigated the impact of proteotoxicity on male sex determination. We have established that enhancing male sex determination in hermaphrodites leads to increased longevity, elevated protein ubiquitination, and reduced UPR^ER and protein aggregation. Conversely, reducing male sex determination in males decreases longevity, diminishes protein ubiquitination, and increases UPR^ER and protein aggregation. Our results also reveal that enhancing male sex determination in *daf-18(D137A)* hermaphrodites by knocking down *tra-3* activates proteasome activity (Fig. 9B). On the other hand, reducing male sex determination in *daf-18(D137A)* males by knocking down *fem-2* decreases proteasome activity (Fig. 9C). Furthermore, the percentage of paralyzed worms and A-beta aggregation in *daf-18(D137A);dvIs2* hermaphrodites is reduced by knocking down *tra-3* (Fig. 9D,F). In contrast, the paralysis rate and A-beta aggregation in *daf-18(D137A);dvIs2* males can be increased by *fem-2* RNAi or MG132 treatment (Fig. 9E,G). In the case of aggregated Alpha-synuclein in *daf-18(D137A);pkIs2386* hermaphrodites, its levels are decreased by *tra-3* RNAi (Fig. 9H). However, aggregated Alpha-synuclein levels in *daf-18(D137A);pkIs2386* males can be increased by *fem-2* RNAi or MG132 treatment (Fig. 9I). Lastly, the motility of *daf-18(D137A);pkIs2386* hermaphrodites is enhanced by *tra-3* RNAi (Fig. 9J). In contrast, the motility of *daf-18(D137A);pkIs2386* males can be reduced by *fem-2* RNAi or MG132 treatment (Fig. 9K).

Collectively, all of these studies provide further confirmation that male sex determination is activated in *daf-18*-deficient worms to maintain lower proteotoxicity through the regulation of *unc-23*, thereby ensuring a survival advantage when compared to hermaphrodites.

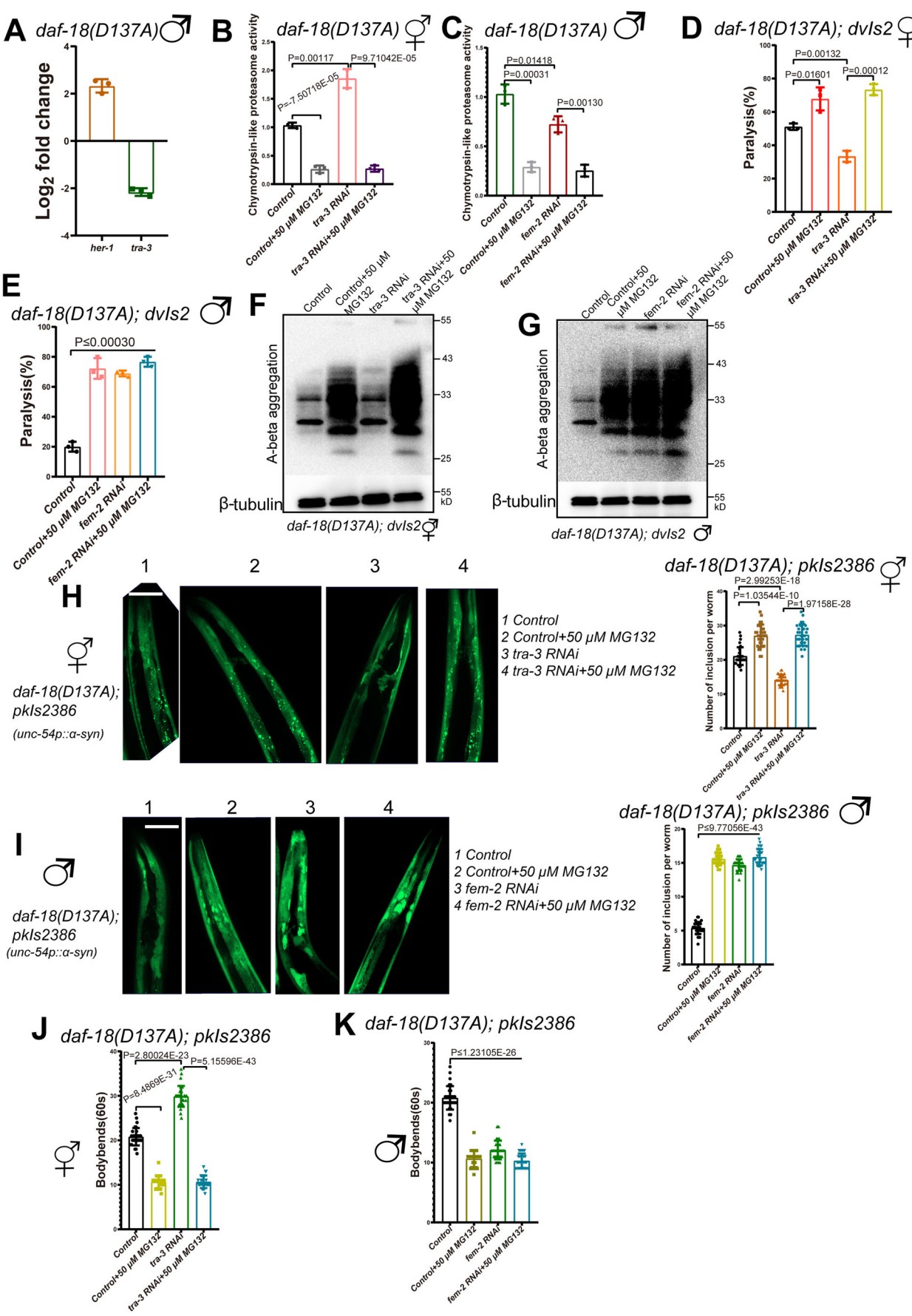

**Figure 9. Enhancing male sex determination activates UPS activity and reduces proteotoxic proteins.**

(A) Log2 fold changes of *her-1* and *tra-3* expression in *daf-18(D137A)* males. The data show the average of three independent repeats, and the error bars show the standard deviations. Chymotrypsin-like proteasome activity (relative slope to control) was assessed in *daf-18(D137A)* hermaphrodites (B) and males (C) fed with L4440 (Control), *fem-2*, or *tra-3* RNAi. The data show the average of three independent repeats, and the error bars show the standard deviations. The *P* value was determined by using a two-tailed *t* test (*P* values; B, *control* hermaphrodites vs *control* + 50 μM MG132 hermaphrodites: 7.50718E-05; *control* hermaphrodites vs *tra-3* RNAi hermaphrodites: 0.00117; *tra-3* RNAi hermaphrodites vs *tra-3* RNAi + 50 μM MG132 hermaphrodites: 9.71042E-05; C, *control* males vs *control* + 50 μM MG132 males: 0.00031; *control* males vs *fem-2* RNAi males: 0.01418; *fem-2* RNAi males vs *fem-2* RNAi + 50 μM MG132 males: 0.00130). Percentage of paralysis in day 5 *daf-18(D137A);dvIs2 [Punc-54::human A-beta 3-42; pRF4 (rol-6(su1006)]* hermaphrodites (D) and males (E) fed with L4440 (Control), *fem-2*, or *tra-3* RNAi. The data show the average of three independent repeats, and the error bars show the standard deviations. The *P* value was determined by using a two-tailed *t* test (*P* values; D, *control* hermaphrodites vs *control* + 50 μM MG132 hermaphrodites: 0.01601; *control* hermaphrodites vs *tra-3* RNAi hermaphrodites: 0.00132; *tra-3* RNAi hermaphrodites vs *tra-3* RNAi + 50 μM MG132 hermaphrodites: 0.00012; E, *control* males vs *control* + 50 μM MG132 males: 0.00030; *control* males vs *fem-2* RNAi males: 2.52639E-05, shown as ≤0.00030; *fem-2* RNAi males vs *fem-2* RNAi + 50 μM MG132 males: 3.14429E-05, shown as ≤0.00030). Western blots of A-beta aggregation in day 5 *daf-18(D137A);dvIs2 [Punc-54::human A-beta 3-42; pRF4 (rol-6(su1006)]* hermaphrodites (F) and males (G) fed with L4440 (Control), *fem-2*, or *tra-3* RNAi. Each immunoblot shows representative of three times independent experiments. Alpha-synuclein-YFP expression in the head region and the average number of inclusions per animal between the tip of the nose and the pharyngeal bulb of day 5 *daf-18(D137A);pkIs2386 [unc-54p::α-syn::YFP]* hermaphrodites (H) and males (I) fed with L4440 (Control), *fem-2*, or *tra-3* RNAi. Scale bar: 50 μm. The data shows the values of all samples from three replicates, with error bars representing the averages and standard deviations. The *P* value was determined by using a two-tailed *t* test (*P* values; H, *control* hermaphrodites vs *control* + 50 μM MG132 hermaphrodites: 1.03544E-10; *control* hermaphrodites vs *tra-3* RNAi hermaphrodites: 2.99253E-18; *tra-3* RNAi hermaphrodites vs *tra-3* RNAi + 50 μM MG132 hermaphrodites: 1.97158E-28; I, *control* males vs *control* + 50 μM MG132 males: 1.37243E-45, shown as ≤9.77056E-43; *control* males vs *fem-2* RNAi males: 1.48062E-44, shown as ≤9.77056E-43; *control* males vs *fem-2* RNAi + 50 μM MG132 males: 9.77056E-43).The mobility of day 5 *daf-18(D137A);pkIs2386 [unc-54p::α-syn::YFP]* hermaphrodites (J) and males (K) fed with L4440 (Control), *fem-2*, or *tra-3* RNAi was assessed using thrashing experiments. Animals that were alive but unable to swim were scored as paralyzed. Motility assays were carried out by counting body bends over a 30-s interval in M9 at day 5 adulthood. Sample size 30. The data shows the values of all samples from three replicates, with error bars representing the averages and standard deviations. *P* values were determined by using two-tailed *t*-test (*P* values; J, *control* hermaphrodites vs *control* + 50 μM MG132 hermaphrodites: 8.4869E-31; *control* hermaphrodites vs *tra-3* RNAi hermaphrodites: 2.80024E-23; *tra-3* RNAi hermaphrodites vs *tra-3* RNAi + 50 μM MG132 hermaphrodites: 5.15596E-43; K, *control* males vs *control* + 50 μM MG132 males: 6.81603E-32, shown as ≤1.23105E-26; *control* males vs *fem-2* RNAi males: 1.23105E-26; *control* males vs *fem-2* RNAi + 50 μM MG132 males: 8.9218E-33, shown as ≤1.23105E-26). Worms carried the *him-5(e1490)* mutation to ensure sufficient males were produced. Source data are available online for this figure.

## Discussion

We found sex-specific differences in response to loss of the autosomal-based tumor suppressor *daf-18/PTEN*. Our results suggest that hermaphrodite worms are more affected by the loss of *daf-18* than male worms, as they experience a significant reduction in adult lifespan. This could be due to an increase in unfolded protein content and a weakened ability of the proteasomal protein degradation system to handle denatured unfolded proteins. When males lose *daf-18*, they can potentially evoke UNC-23/NEF-regulated ubiquitination to reduce the burden of unfolded proteins induced by loss of *daf-18*.

DAF-18 protein phosphatase plays a crucial role in maintaining ER-related protein homeostasis. DAF-18 protein phosphatase deficiency resulted in the failure to dephosphorylate C18E9.2, a predicted ER membrane-bound protein. This led to a heightened unfolded protein response in the ER and global protein aggregation in hermaphrodites. C18E9.2 is predicated to be a homology of human SEC62, which is known to interact with ribosomes and is a well-characterized component of the translocation machinery responsible for guiding newly synthesized precursor polypeptides into the ER (Conti et al, 2015). In the ER, post-translational freshly unfolded proteins are properly folded and guided by folding enzymes and chaperones (Tannous et al, 2015). Misfolded polypeptides are eventually selected, translocated across the ER membrane into the cytosol, and degraded by proteasomes through ERAD (Tannous et al, 2015). The activation of UPR^ER in *daf-18(ok480)* worms may result from ER-resident folding failures when serine 301 on C18E9.2/SEC62 is phosphorylated. Subsequently, secreted unfolded proteins are selected and ubiquitinated under the control of UNC-23/NEF. The lack of dephosphorylation of C18E9.2 caused unfolded proteins to reduce the adult lifespan of worms; however, DAF-18 protein phosphatase deficiency may evoke male sex determination to enhance male survival. We also found that the expression of *her-1* and *unc-23*

was upregulated in *daf-18(D137A);C18E9.2(S301A)* males (Appendix Fig. S11), suggesting that the ability of male sex determination and UNC-23/NEF chaperone to select unfolded protein is enhanced whether the unfolded protein is produced or not. Despite years of research, the function of SEC62 in ER-related protein aggregation remains unclear. Our study may shed light on the role of SEC62 in aging biology.

In *daf-18(ok480)* males, secreted unfolded proteins from the ER are selected and ubiquitinated with the assistance of UNC-23/NEF. This unfolded protein ubiquitination is believed to stimulate proteasome activity, facilitating the degradation of aggregated proteins and preventing proteotoxicity. This explanation clarifies how males maintain a survival advantage. NEF collaborates with other chaperones to form a protein disaggregation machinery that solubilizes amyloid-like aggregates, which can result from acute stress, protein misfolding, and the decline or collapse of protein quality control during aging and disease (Hipp et al, 2014). This aligns with our findings that proteotoxicity induced by A-beta and Alpha-synuclein can be alleviated by UNC-23/NEF regulation, activated through male sex determination in *daf-18(ok480)* males.

According to our results, we speculate that NEF-related protein homeostasis, which is enhanced in *daf-18(ok480)* males, may be regulated by the male-specific sex determination pathway. The previous findings that male pheromones and related pathways help regulate survival enhancement and cancer amelioration are also consistent with our report (Hickey et al, 2021); we propose that male sex determination may play the pivotal role. Human females and hermaphrodite worms live longer than males, and human females have increased cancer incidence (Bronikowski et al, 2022). However, upon loss of *daf-18/PTEN*, this advantage may be reversed; the *daf-18/PTEN*-deficient males have the ability to maintain proteostasis to sustain survival and probably reduce cancer risk. Our finding is consistent with clinical reports that PTEN mutations are more frequently found in female patients with

cancers such as breast and endometrial cancers (11% within the general population), while males with PTEN mutations have a lower (<1% within the general population) chance of developing cancers than females (Blumenthal and Dennis, 2008).

A previous study proposed that enhancing male sex determination signaling could regulate IIS to extend adult lifespan of hermaphrodites (Hotzi et al, 2018). In our case, IIS is also need to extend the adult lifespan of both males and hermaphrodites. However, our RNA sequencing analysis indicates that IIS may not play a crucial role in controlling male survival advantage. Specifically, the analysis showed similar changes in IIS and the target genes of DAF-16 between males and hermaphrodites, indicating that the genes responsible for the sex-biased survival are unlikely to be in the IIS pathway after *daf-18* loss. We found that the transcription factors controlled by IIS did not play roles in male sex bias survival. The survival advantage of males in the absence of *daf-18* may depend on the protein phosphatase activity of DAF-18, rather than lipid phosphatase activity, which is more critical factor in IIS. The protein phosphatase activity of DAF-18 was reported to alter some lipid phosphatase activity (Chen et al, 2022; Wittes and Greenwald, 2022); however, the ER-related protein homeostasis regulated by DAF-18 protein phosphatase may be the main reason causing sex bias survival. However, how the protein phosphatase activity of DAF-18 regulates the male sex determination are still needed to be addressed.

# Methods

### Reagents and tools table

| Reagent/resource | Reference or source | Identifier or catalog number |
|---|---|---|
| **Experimental models** | | |
| *pkIs2386* | CGC | NL5901 |
| *dvIs2* | CGC | CL2006 |
| *dvIs70* | CGC | FB217 |
| *zcIs4* | CGC | SJ4005 |
| *zcIs14* | CGC | SJ4103 |
| *zcIs13* | CGC | SJ4100 |
| *ccIs4251* | CGC | SD1643 |
| **Recombinant DNA** | | |
| Punc-23::unc-23:unc-54utr | This study | |
| Pdaf-18::daf-18::unc-54utr | Zheng et al, 2018b | |
| Pdaf-18::daf-18 G174E::unc-54utr | Zheng et al, 2018b | |
| Pdaf-18::daf-18 D137A::unc-54utr | Zheng et al, 2018b | |
| Pdaf-18::daf-18 D137A::unc-54utr | Zheng et al, 2018b | |
| L2528 | Addgene | Plasmid #1606 |
| pJRK248 | Addgene | Plasmid #173755 |

| Reagent/resource | Reference or source | Identifier or catalog number |
|---|---|---|
| **Antibodies** | | |
| Anti-ubiquitin (P4D1) mouse mAb | CST, | 3936S |
| K48 antibody | Abcam | ab140601 |
| Anti-β-tubulin | Abways | AB0039 |
| Goat anti-rabbit IgG (H + L) (peroxidase/HRP conjugated) | Elabscience | E-AB-1003) |
| HRP-labeled goat anti-mouse IgG (H + L) | Beyotime | A0216 |
| β-amyloid 1–16 (6E10) | PTM Bio | PTM-20007 |
| **Oligonucleotides and other sequence-based reagents** | | |
| *her-1* forward | ACCCCAATTCGATGTGGTTA | OSQZ100 f |
| *fem-2* forward | TCTCGACGAACGAATGACTG | OSQZ103 f |
| *fem-3* forward | TTGTACGGCCTTTTCCAATC | OSQZ104 f |
| *pek-1* forward | GCAGCGTGAACAACACAACT | OSQZ41 f |
| *skr-8* forward | CCAACATGGCTATCGGAAAA | OSQZ158 f |
| *F44E5.4* forward | TGTTGCTGGTTGATGTTGCT | OSQZ39 f |
| *F44E5.5* forward | GGATGTGTTGCTGGTTGATG | OSQZ40 f |
| *hsp-70* forward | AGCCCGTTGTTGAGGTTGAA | OSQZ23 f |
| *hsp-110* forward | GCAATTGCTCTAGCCTACGG | OSQZ38 f |
| *unc-23* forward | TGATCAGAAACGCATCAAGC | OSQZ42 f |
| *tra-3* forward | ACATACAGCCATGCATCCAA | OSQZ101 f |
| *tra-2* forward | TTCATTGAATGGATGCAGGA | OSQZ159 f |
| *sel-1* forward | GCTCAATTGGGACTCGGACA | OSQZ160 f |
| *sel-11* forward | CCTCAACATCCTCAACCGCT | OSQZ161 f |
| *atf-6* forward | TGTTCAGCCCTTGATGCCAT | OSQZ162 f |
| *xbp-1* forward | TCGCAGCCCAAAATGCTAGA | OSQZ163 f |
| *ire-1* forward | GTTCCGGAGAGGCTGTCTTC | OSQZ164 f |
| *cdc-42* forward | CTGCTGGACAGGAAGATTACG | OSQZ3 f |
| *her-1* reverse | TGCAAACAACACATTTTGAAGA | OSQZ100 r |
| *fem-2* reverse | CTGATCCATGTCGATTGCAC | OSQZ103 r |
| *fem-3* reverse | GTGGATAAAAAGCGGCGATA | OSQZ104 r |
| *pek-1* reverse | TGTTGAAAGCTCTGGCAATG | OSQZ41 r |
| *skr-8* reverse | TCTCCTTTGCAGCTCTCTCC | OSQZ158 r |
| *F44E5.4* reverse | TGTTTTGCAGGCTTTTGTTG | OSQZ39 r |
| *F44E5.5* reverse | TGTTTTGCAGGCTTTTGTTG | OSQZ40 r |
| *hsp-70* reverse | CCCGTACAGAATGCCCAAGT | OSQZ23 r |
| *hsp-110* reverse | AAGCAACCAATGAAGCCTGT | OSQZ38 r |
| *unc-23* reverse | CCATCAAATCTGCTTTCGTTC | OSQZ42 r |
| *tra-3* reverse | TGTCAACGGTGAAAAATGGA | OSQZ101 r |
| *tra-2* reverse | ACAGGACAATTTCCGTTTGC | OSQZ159 r |
| *sel-1* reverse | CACTTCCTGACTCAGCAGCA | OSQZ160 r |
| *sel-11* reverse | CCTAGAAGACGTGCTAGGCG | OSQZ161 r |
| *atf-6* reverse | TCGGACACTTGTCGAACCAG | OSQZ162 r |
| *xbp-1* reverse | AAGACGTTCGTTTTCAGCGC | OSQZ163 r |

| Reagent/resource | Reference or source | Identifier or catalog number |
|---|---|---|
| *ire-1 reverse* | GACAGGTTTTGGTGCTCGTG | OSQZ164 r |
| *cdc-42 reverse* | CTCGGACATTCTCGAATGAAG | OSQZ3 r |
| **Chemicals, enzymes and other reagents** | | |
| Ampicillin | Sangon Biotech | 69-52-3 |
| Tetracycline | Sangon Biotech | 64-75-5 |
| Isopropyl-B-D-thiogalactopyranoside | Sigma | 15502 |
| RNAiso Plus | Takara | 9109 |
| Phusion High-Fidelity PCR Master Mix | Thermo Scientific | F531L |
| High Capacity cDNA Reverse Transcription Kit | Applied Biosystems | 4368814 |
| Power SYBR Green PCR Master Mix | Applied Biosystems | 4368577 |
| Sodium azide | Sigma | 26628-22-8 |
| GenElute Gel Extraction Kit | Sigma | NA1111 |
| Quick Ligation™ Kit | NEB | M2200L |
| PROTEOSTAT Protein Aggregation Assay | Enzo Life Sciences | ENZ-51023-KP002 |
| Z-Gly-Gly-Leu-AMC | Sigma | SCP0225 |
| **Software** | | |
| GraphPad Prism v10 | | GraphPad Software |
| ImageJ | | National institutes of health |
| SPSS | | IBM statistics 21 |

## Strains and maintenance

Worms were grown and maintained on standard nematode growth medium (NGM) at 20 °C as previously described (Brenner, 1974), unless otherwise indicated. Strains used in this study: wild-type: N2, RB712: *daf-18(ok480)*, CB4088: *him-5(e1490)*, SJ4103: *zcIs14 [myo-3p::GFP(mit)]*, SD1643: *ccIs4251;stIs10632*, SJ4005: *zcIs4 [hsp-4p::GFP]*, SJ4100: *zcIs13 [hsp-6p::GFP + lin-15(+)]*, FB217: *dvIs70 [Phsp-16.2::gfp; pRF4 (rol-6(su1006)]*, CB3844: *fem-3(e2006)*, CB25: *unc-23(e25)*, DH245: *fem-2(b245)*, MT1446: *her-1(n695)*, CF1038: *daf-16 (mu86)*, EU1: *skn-1(zu67)*, PS3551: *hsf-1 (sy441)*, RB711: *pqm-1 (ok485)*, *daf-18 (mg198)*, CB1375: *daf-18 (e1375)*, NS3227: *daf-18 (nr2037)*, CL2006: *dvIs2 [Punc-54::human A-beta 3-42; pRF4 (rol-6(su1006)]*, NL5901: *pkIs2386 [unc-54p::α-syn::YFP+unc-119(+)]*. Other strains made in our lab: SQZ26: *daf-18(aqz4, D137A)*, SQZ35: *daf-18 (aqz5, G174E)*, SQZ36: *C18E9.2 (aqz6, S301A)*, SQZ37: *C18E9.2 (aqz7, S301E)*. The *him-5(e1490)* mutation was used in this work to generate males.

## Survival and lifespan assays

The survival of worms in the L1 arrest stage was analyzed by a method described previously (Zheng et al, 2018a). In brief, well-maintained mixed-staged worms were collected and lysed by using bleach buffer (5 N NaOH with 1 mL household bleach) to prepare eggs. The eggs were maintained and hatched in sterile M9 and incubated at 20 °C with low-speed rocking to initiate L1 arrest. To properly analyze the sex difference survival of *daf-18*-deficient males and hermaphrodites in L1 diapause status, the mixed worms were incubated in M9 in the presence of 0.08% ethanol, which can increase worm lifespan (Fukuyama et al, 2012). A 50–100 μL aliquot of M9 containing a minimum of 100 L1 arrested worms was transferred to an NGM plate seeded with OP50 every day. The recovered L1 diapause worms were cultured for 3 days to allow them to grow to young adults, and then the percentage of surviving males or hermaphrodites was calculated (Fig. EV1B).

For adult lifespan analysis, synchronized L3-L4 hermaphrodites or males were randomly picked from maintenance plates. The worms were assessed for death or survival every day.

## RNAi

L4 hermaphroditic or male worms were fed HT115 strains containing an empty control vector (L4440) or expressing a double-stranded target gene RNAi construct (vector, L4440) and were cultured according to a standard protocol described by the company Horizon Discovery and reported previously (Lezzerini et al, 2015). In brief, replicate the control clone and interest RNAi clones from library stock plates onto the LB agar plates containing ampicillin (100 μg/mL) and tetracycline (12.5 μg/mL). using a sterile pin replicator. Inoculate the RNAi clones and controls from the LB plates into LB liquid medium with the same antibiotics to culture overnight at 37 °C on a shaker. The culture of these RNAi colons were seeded onto NGM-ampicillin (100 μg/mL) plates supplemented with 1 mM isopropyl-B-D-thiogalactopyranoside (IPTG) (15502, Sigma) for the induction of the double-stranded RNA. L4 larva were then transferred to the same RNAi plates and cultured at 20 °C for 24 h. The efficiency of RNAi was confirmed by real-time PCR.

## Real-time PCR

Total RNA was extracted from well-maintained worms by using RNAiso Plus (9109, Takara) and converted to cDNA by using a High Capacity cDNA Reverse Transcription Kit (4368814, Applied Biosystems). Real-time PCR experiments were performed using Power SYBR Green PCR Master Mix (4368577, Applied Biosystems) and an ABI 7500 system. Three technical replicates were performed in each reaction. The housekeeping gene *cdc-42* was used as internal control in the experiments and, the results were from at least three biological replicates. All primers used for real time PCR were summarized in Appendix Table S6.

## Quantification of hatched eggs and development into adults

Each NGM plate contained a single worm and was cultured at 20 °C. Each worm was cultured on the plate for 24 h to lay eggs and then transferred onto a new NGM plate every day until it laid no eggs. The plates containing eggs were collected each day and kept at 20 °C to let the eggs hatch and grow to the L4 stage. The male and hermaphroditic progeny of each worm were counted separately.

To assess the progeny number of males, a single young adult control and tested mutant male were cultured with 10 *fem-3* hermaphrodites (this mutation causes these worms to produce only oocytes) at 24 °C. The males were transferred onto new NGM plates seeded with food, and another 10 new young adult *fem-3* hermaphrodites were also transferred into this plate to provide oocytes each day until no progeny were laid. The male and hermaphroditic progeny on this plate were counted each day.

## GFP fluorescence

For fluorescence imaging, synchronized later L4-young adult males and hermaphrodites were anesthetized using an M9 salt solution containing 30 mM sodium azide (26628-22-8, Sigma) and mounted onto 2% agar pads. The animals were then visualized using a Leica M165 FC fluorescence stereomicroscope.

## RNA sequencing

Synchronized young adult males or hermaphrodites were collected. The males and hermaphrodites were separated through picking, and the process needs to be completed from the L3 to young L4 stage to avoid mating. The time window for this picking process is narrow, at least five people to pick for several hours to prepare one sample according to our experience. Usually, 50–100 mg worm pellets were set to do the RNA sequencing, or less worms were picked but the total RNA yielded passed the quantify test (about 50 ng/µL, 35 µL in total). Total RNA quality was measured using a NanoDrop ND-1000 (Agilent Technologies). Library preparation was performed using a KAPA Stranded RNA-Seq Library Prep Kit (Illumina), and the quality was confirmed using an Agilent 2100 Bioanalyzer (Agilent Technologies). Transcriptome alignment and quantification were performed using HISAT2 software. The *C. elegans* genome version WBcel235 was used as the reference.

## Transgenic strain

The *unc-23* gene sequence was amplified from N2 *C. elegans* genomic DNA by using Phusion High-Fidelity PCR Master Mix (F531L, Thermo Fisher). The *unc-23* genomic sequence including the 2 kb UTR sequence upstream ATG was amplified from N2 *C. elegans* genomic DNA by using the primers:

Forward: 5'-aattcccgggGGTTGCCTATCACTGATTGTTTGTA-CAAACACAGACACGAG-3'

Reverse: 5'-aattgctagcCTATTCGCTTTGATCATCCATCAAA TCTGCTTTCGTTC-3'.

The amplified PCR product and the worm expression plasmid L2518 were digested by Nhe1 and Sma1 separately, purified by using the GenElute Gel Extraction Kit (NA1111, Sigma). The purified PCR product and plasmid were mixed in a 3:1 ratio, and linked by using Quick Ligation™ Kit (M2200L, NEB). The *daf-18* rescue plasmids were constructed as previously reported, please see the detail in our previous publishment (Zheng et al, 2018b). All constructions were confirmed by sequencing. The DNA mixtures for injection consisted of 20–50 ng/µL of the interest plasmid and the injection marker (roller) pJRK 248 (2-5 ng/µL, Addgene) was injected into worms using standard microinjection methods (Mello et al, 1991). Each injected strain had at least three stable lines.

## PROTEOSTAT analysis

A PROTEOSTAT Protein Aggregation Assay (Enzo Life Sciences, Inc., ENZ-51023-KP002) (Miyake et al, 2022) was used to monitor total protein aggregation in *C. elegans* samples. In each experiment, *C. elegans* protein was diluted to a consistent final concentration within the recommended concentration range (1 µg/mL to 10 mg/mL) in the test buffer. Samples were then transferred to a 96-well microplate containing 2 µL of the diluted PROTEOSTAT detection reagent in each well. Fluorescence was measured using a fluorescence microplate reader (BMG LABTECH; CLARIOstar-plus) with an excitation wavelength of 550 nm and an emission wavelength of 600 nm.

## Western blot analysis

A previously reported standard method was used to assess ubiquiti-nated proteins by western blotting (Koyuncu et al, 2021). Total protein (20 µg) was separated by SDS–PAGE, transferred to polyvinylidene difluoride membranes (Millipore) and subjected to immunoblotting. Western blot analysis was performed with an anti-ubiquitin (P4D1) mouse mAb (CST, 1:200, 3936S), an anti-ubiquitin linkage-specific K48 antibody (Abcam, EP8589, 1:200, ab140601), an anti-β-tubulin antibody (Abways, 1:5000, AB0039), goat anti-rabbit IgG (H + L) (peroxidase/HRP conjugated) (Elabscience, 1:1000, E-AB-1003) and HRP-labeled goat anti-mouse IgG (H + L) (Beyotime, 1:1000, A0216), β-amyloid 1–16 (6E10) (1:1000, PTM-20007; PTM Bio).

## 26S proteasome fluorogenic peptidase assays

The 26S proteasome activity assays were performed as previously described (Vilchez et al, 2012b). 25 µg of total protein lysate was transferred to a 96-well microtiter plate (BD Falcon), and a fluorogenic substrate was added during each test. Z-Gly-Gly-Leu-AMC (SCP0225, Sigma) was used to measure the chymotrypsin-like activity of the proteasome. Fluorescence (380-nm excitation, 460-nm emission) was monitored using a fluorometer (Infinite E Plex) every 5 min for 1 h at 25 °C. The relative slopes to control of fluorescence curves were used to demonstrate changes in proteasome activity.

## Alpha-synuclein inclusion measurement

The previously method was used (Kim et al, 2008). In brief, the formation of *a-synuclein* inclusions containing aggregated *a-synuclein* resembling a critical pathological feature in living *C. elegans* can be monitored by using a confocal microscope. The day 5 of adulthood worms with *pkIs2386* [*unc-54p::alpha-synuclei-n::YFP + unc-119(+)*] were mounted on 2% agarose pads containing 40 mM NaN3 as an aesthetic on glass microscope slides. Confocal images were captured by a Zeiss 880 inverted laser scanning confocal microscope with an x40 oil immersion lens. For quantification of the number of inclusions all foci between the nose and pharyngeal bulb were analyzed. Measurements on inclusions were performed using ImageJ software.

## Motility

The motility of the worms was analyzed using a thrashing assay as previously described (Volovik et al, 2014). In brief, a day 5 single

worm expressing Alpha- synuclein (*pkIs2386*) was transferred into a drop of M9 placed on a coverslip. The number of body bends was counted per 30 s under a stereoscope.

## Paralysis assay

This was tested using a protocol previously published (Volovik et al, 2014). Synchronized worms expressing A-beta (*dvIs2*) were placed on NGM plates or RNAi plates seeded with the control and test bacteria. They were then transferred onto fresh plates every two days until day 5.

## Post-translational modification analysis (protein phosphoromics)

Synchronized young adult N2 and *daf-18(D137A)* worms were used to perform this analysis. Approximately 50 mg of worm pellets per test were prepared for protein extraction, with 1% phosphatase inhibitor added for phosphorylation. Total proteins were extracted using lysis buffer (8 M urea, 1% protease inhibitor cocktail), and protein concentration was determined using a BCA kit. Samples were treated with 20% TCA, vortexed, and incubated at 4 °C for 2 h, then centrifuged at 4500 g for 5 min. The precipitate was washed in cold acetone, dried briefly, and redissolved in 200 mM TEAB with ultrasonic dispersion. Trypsin digestion occurred at a 1:50 mass ratio overnight, followed by reduction with 5 mM dithiothreitol and alkylation with 11 mM iodoacetamide. Peptides were desalted, enriched using IMAC microspheres, and eluted with 10% NH4OH, then lyophilized for LC-MS/MS analysis. The protein quantitative testing and phosphorylation modification was performed in PTM bio (Hangzhou, China).

## Statistical analyses

All graphed data are presented as the average of at least 3 biological replicate experiments performed in triplicate technical repeats. Error bars show the standard deviations. The RNA-seq data were analyzed by using the Ballgown package of the R project. The raw data of RNA-seq was submitted to GEO (GSE222447). Statistical calculations were performed using GraphPad Prism 10 (GraphPad Software) and SPSS (IBM Statistics 21). The mean fluorescence intensity and bands of western blotting were quantified using ImageJ. The lifespan curves were generated by using the Kaplan–Meier method. The significance of the difference in the overall survival rate was determined using the log-rank test. The relative expression levels of genes were determined using the $2^{-\Delta\Delta CT}$ method. The differences between two groups were analyzed using two-tailed Student's $t$ test, and a $P$ value $< 0.05$ was indicative of statistical significance. Statistical significance values are indicated as follows: $*P < 0.05$, $**P < 0.01$, $***P < 0.001$, and n.s.: not significant. Further details are provided in the figure legends.

## Data availability

Source data are provided with this paper. RNA sequencing data that support the findings of this study have been deposited in GEO with the accession code: GSE222447 (https://www.ncbi.nlm.nih.gov/geo/query/acc.cgi?acc=GSE222447).

The source data of this paper are collected in the following database record: biostudies:S-SCDT-10_1038-S44319-025-00368-x.

## Peer review information

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

## Acknowledgements

We are grateful to Caenorhabditis Genomic Center for providing strains. We thank Professor Mengqiu Dong (NIBS) and Professor Ian Chinsang (Queen's University) for sharing strains. This work was supported by Henan University (Yellow River Scholar Fund, CX3050A0250205), The Youth Promotion Project of Zhongzhou Laboratory for Integrative Biology (CXK2024TS0103).

## Author contributions

**Zhi Qu**: Conceptualization; Resources; Investigation; Writing—review and editing. **Lu Zhang**: Formal analysis; Investigation; Methodology. **Xue Yin**: Software; Investigation. **Fangzhou Dai**: Software; Investigation; Visualization. **Wei Huang**: Software; Validation; Methodology. **Yutong Zhang**: Investigation; Visualization. **Dongyang Ran**: Software; Visualization. **Shanqing Zheng**: Conceptualization; Resources; Data curation; Formal analysis; Supervision; Funding acquisition; Validation; Visualization; Methodology; Writing—original draft; Project administration; Writing—review and editing.

Source data underlying figure panels in this paper may have individual authorship assigned. Where available, figure panel/source data authorship is listed in the following database record: biostudies:S-SCDT-10_1038-S44319-025-00368-x.

## Disclosure and competing interests statement

The authors declare no competing interests.

# Expanded View Figures

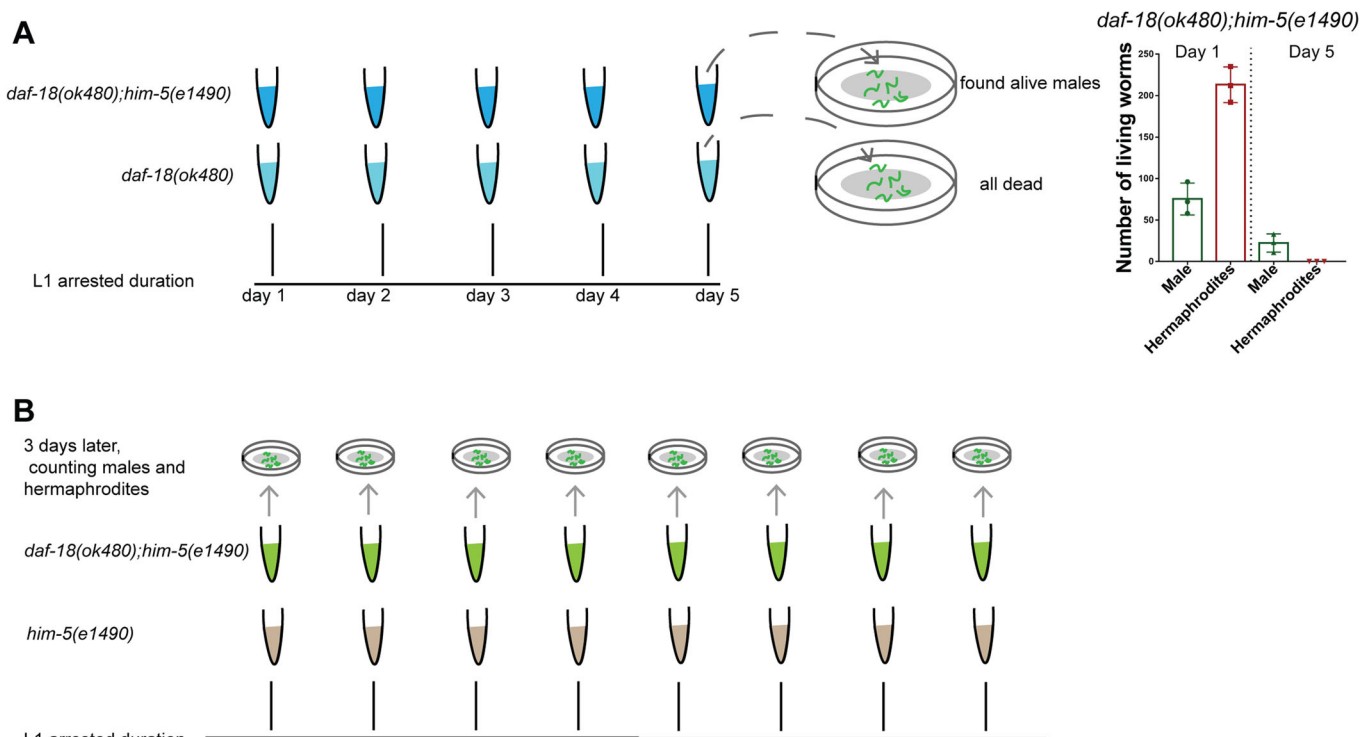

**Figure EV1. *daf-18(ok480)* males live longer than hermaphrodites in L1 arrest stage.**

(A) L1 arrested *daf-18(ok480)* hermaphrodites normally live about 4 days in M9, males were found alive after day 4. The surviving L1 arrested worms were recovered into NGM plates seeded with OP50, and checked after 3 days culturing at 20°C. 100 μL liquid with L1 arrested worms were recovered each repeat. (B) *him-5(e1490)* and *daf-18(ok480);him-5(e1490)* worms were maintained in M9 added with 0.08% (v/v) ethanol, transferred 50-100 μL liquid with L1 arrested worms (more than 100) every day into NGM plates seeded with OP50, and checked 3 days after culturing at 20 °C. The percentages of males and hermaphrodites were calculated every day. The experiment was repeated three times independently. The data show the average of three independent repeats, and the error bars show the standard deviations.

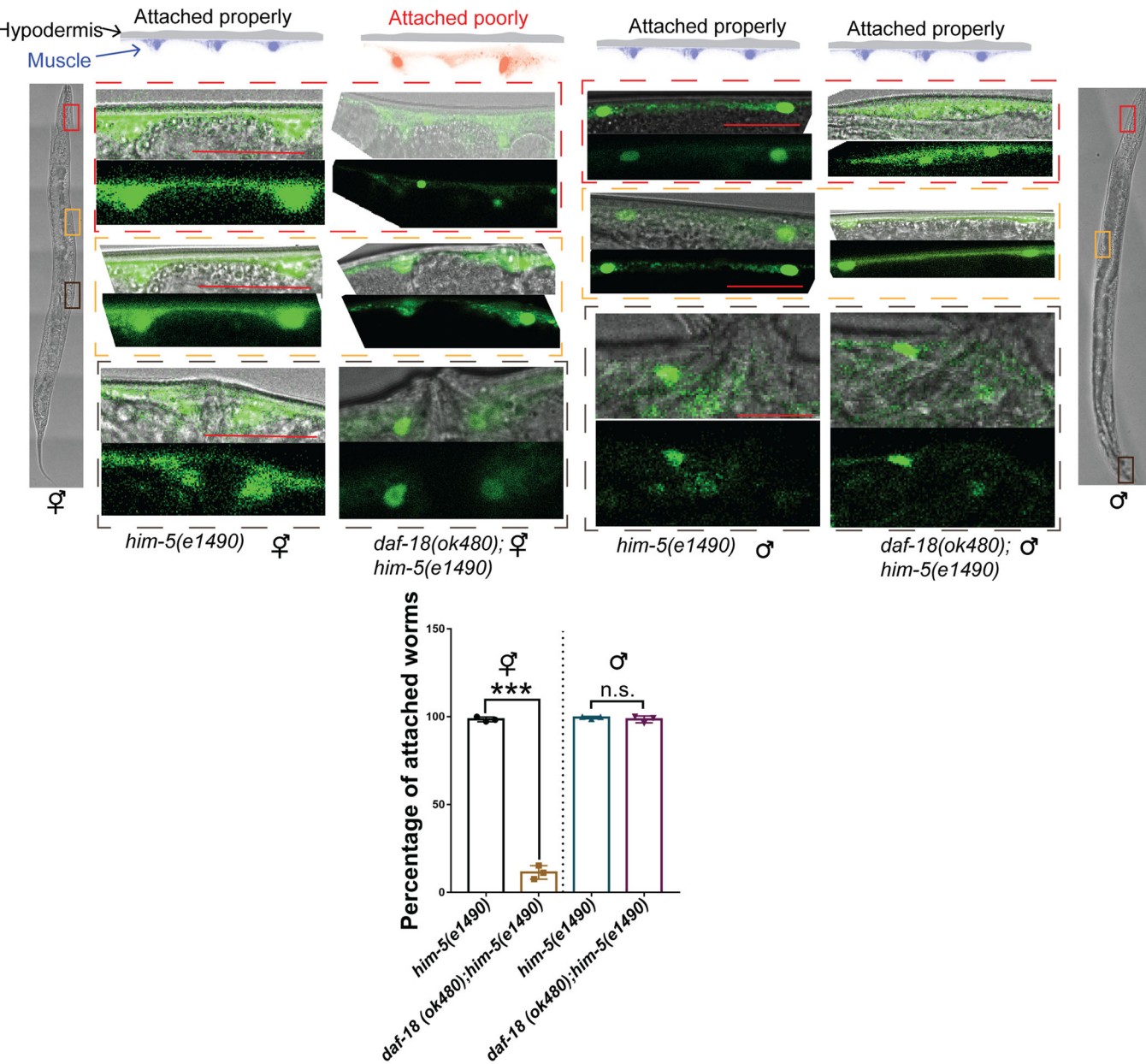

**Figure EV2.** *daf-18(ok480)* hermaphrodites reduce the muscle attachment with hypodermis.

*him-5(e1490)* and *daf-18(ok480);him-5(e1490)* hermaphrodites were checked for muscle attachment with hypodermis at head, vulva and worm body. The status of attachment at all these three points was counted as "Attached properly". *him-5(e1490)* and *daf-18(ok480);him-5(e1490)* males were checked for muscle attachment with hypodermis at head, tail and worm body. The status of males at all these three points was counted as "Attached properly". The experiment was repeated three times independently, with sample size (*n*) = 60 for each mutant. The data show the average of three independent repeats, and the error bars show the standard deviations. *P* value was determined by using two-tailed *t-test*. ***P = 3.11987E-06, n.s.: no significant difference=0.42777. Red scale bar: 50 μm. As *unc-23* is responsible for the proper attachment of hypodermis and muscles, the downregulated *unc-23* in *daf-18* hermaphrodites resulted in detachment. However, males had no significant change, which also suggest males may up-regulate *unc-23* expression to defense against the loss of *daf-18*.

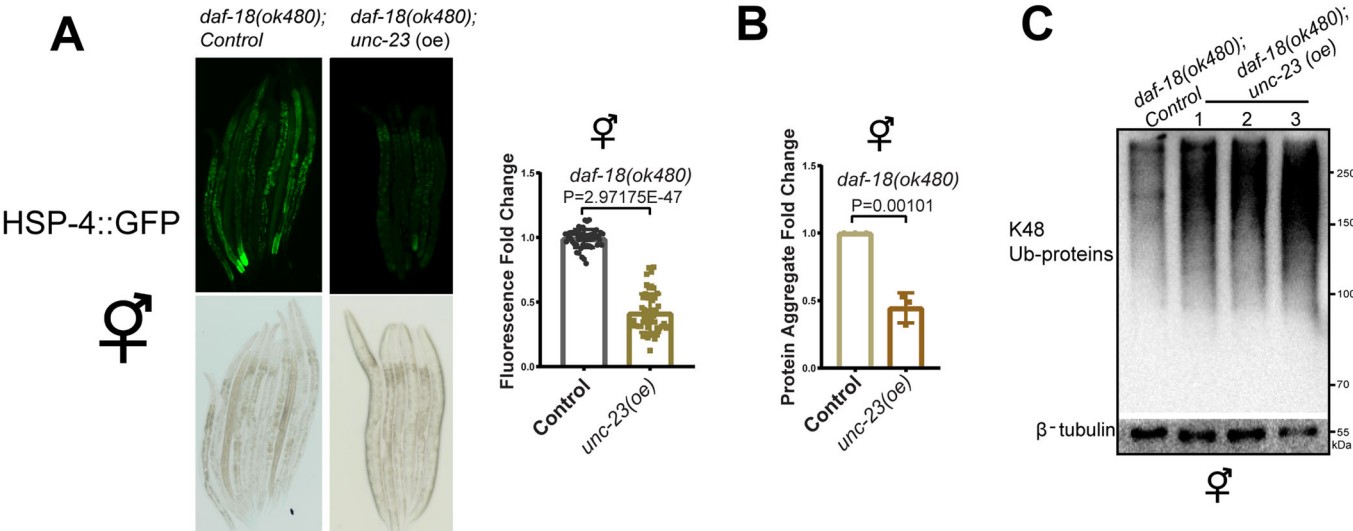

**Figure EV3.  *unc-23* overexpression in *daf-18(ok480)* hermaphrodites rescued the protein homeostasis.**

(A) The unfold protein marker *hsp-4p*::GFP in hermaphrodites. Each experiment set has three independent repeats, sample size (n)=54. Control: the transgenic injection strains with empty expression vector L2528. The data shows the values of all samples from three replicates, with error bars representing the averages and standard deviations. *P* values were determined by using two-tailed *t-test* (*P* value: 2.97175E-47). Scale bar: 200 μm. (B) The total level of protein aggregation in hermaphrodites analyzed by using PROTEOSTAT Protein Aggregation Assay. The fold changes of protein aggregation in hermaphrodites of three independent experiments. Control: the transgenic injection strains with empty expression vector L2528. The data show the average of three independent repeats, and the error bars show the standard deviations. *P* value was determined by using two-tailed *t-test* (*P* value: 0.00101). (C) The k48 linked protein ubiquitination of hermaphrodites. Control: the transgenic injection strains with empty expression vector L2528. Each immunoblot shows representative of three times independent experiments, all the data details summarized in Appendix Table S5.

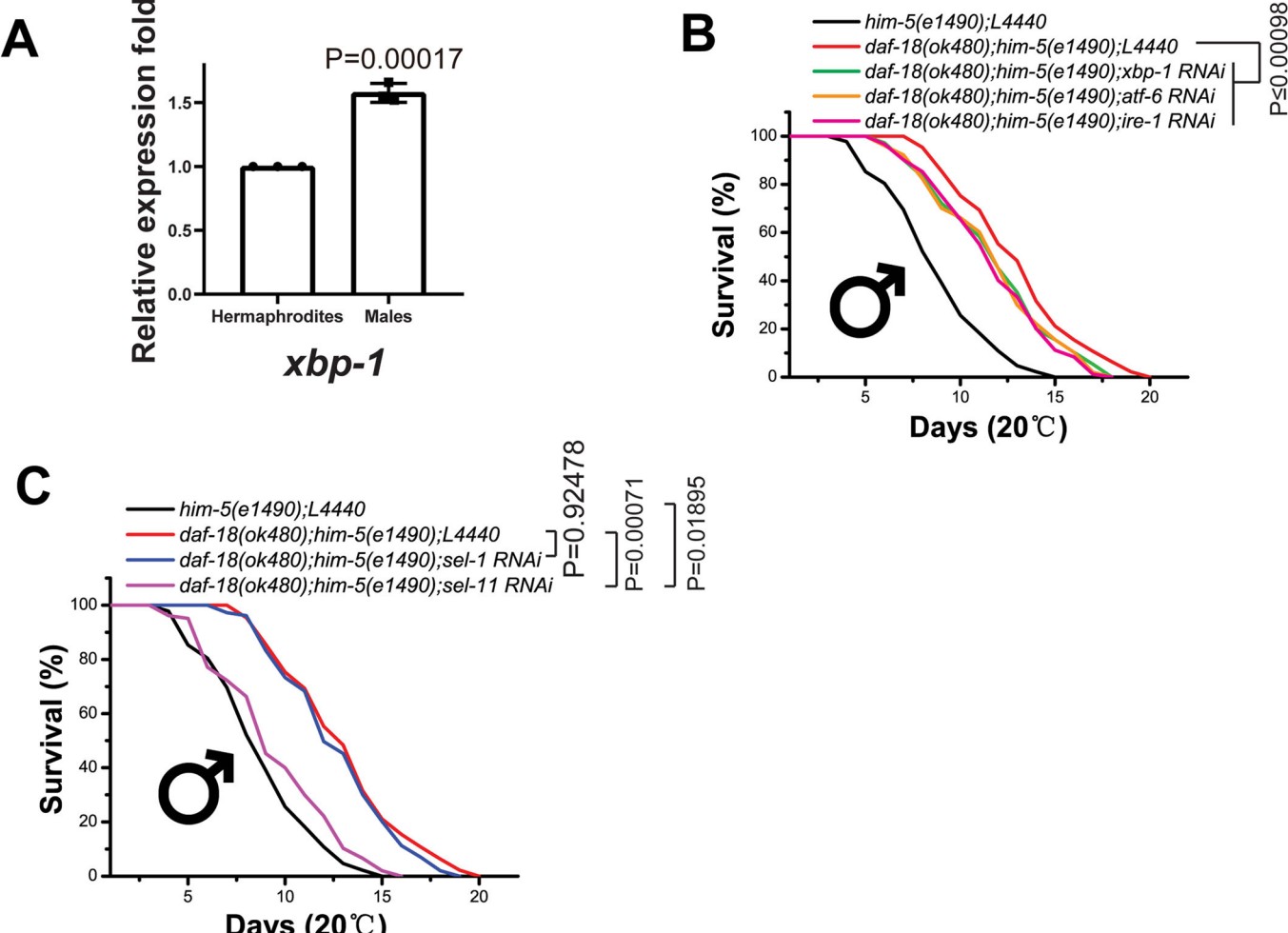

**Figure EV4. The HRD1/SEL-11 and XBP-1/ERAD may be involved in regulating the male survival advantage when *daf-18* is lost.**

(A) The expression of *xbp-1* in *daf-18(ok480)* males is higher than that in hermaphrodites. The data show the average of three independent repeats, and the error bars show the standard deviations. *P* value was determined by using two-tailed *t-test* (*P* value: 0.00017). (B) Knocking down *xbp-1* (*n* = 65), *atf-6* (*n* = 102) and *ire-1* (*n* = 69) can reduce the adult lifespan of *daf-18(ok480)* males. (C) Knocking down *sel-11* (*n* = 97), not *sel-1* (*n* = 95), significantly change the adult lifespan of *daf-18(ok480)* males. Control: RNAi control clones containing the empty vector L4440. Each lifespan experiment set was repeated three times. The mean survival rates were calculated using the Kaplan–Meier method, and *P* values were determined by using the log rank test (*P* values; **B**, *daf-18(ok480);him-5(e1490);*L4440 males vs *xbp-1 RNAi* males: 0.00027; *daf-18(ok480);him-5(e1490);*L4440 males vs *atf-6 RNAi* males: 0.00089; *daf-18(ok480);him-5(e1490);*L4440 males vs *ire-1 RNAi* males: 0.00075; **C**, *daf-18(ok480);him-5(e1490);*L4440 males vs *sel-1 RNAi* males: 0.92478; *daf-18(ok480);him-5(e1490);*L4440 males vs *sel-11 RNAi* males: 0.00071; *him-5(e1490);*L4440 males vs *daf-18(ok480);him-5(e1490);sel-11 RNAi* males: 0.01895). All the lifespan data are summarized in Appendix Table S4.

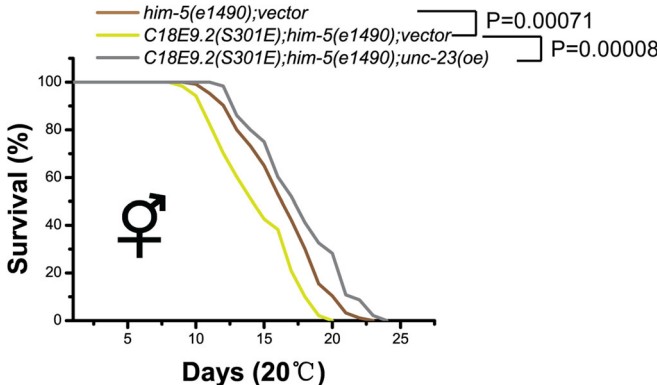

**Figure EV5. Overexpression of *unc-23* can extend the adult lifespan of *C18E9.2(S301E)* hermaphrodites.**

The lifespan of *C18E9.2(S301E)* can be affected by overexpression of *unc-23*. Sample size (*n*) = 68 (*him-5 vector*), 74 (*C18E9.2 vector*), 63 (*unc-23 oe*). Each lifespan experiment set was repeated three times. The mean survival rates were calculated using the Kaplan–Meier method, and *P* values were determined by using the log rank test (*P* values; *him-5(e1490);vector* hermaphrodites *vs C18E9.2(S301E);him-5(e1490);vector* hermaphrodites: 0.00071; *C18E9.2(S301E);-him-5(e1490);vector* hermaphrodites vs *C18E9.2(S301E);him-5(e1490);unc-23(oe)* hermaphrodites: 0.00008). All the lifespan data are summarized in Appendix Table S4.

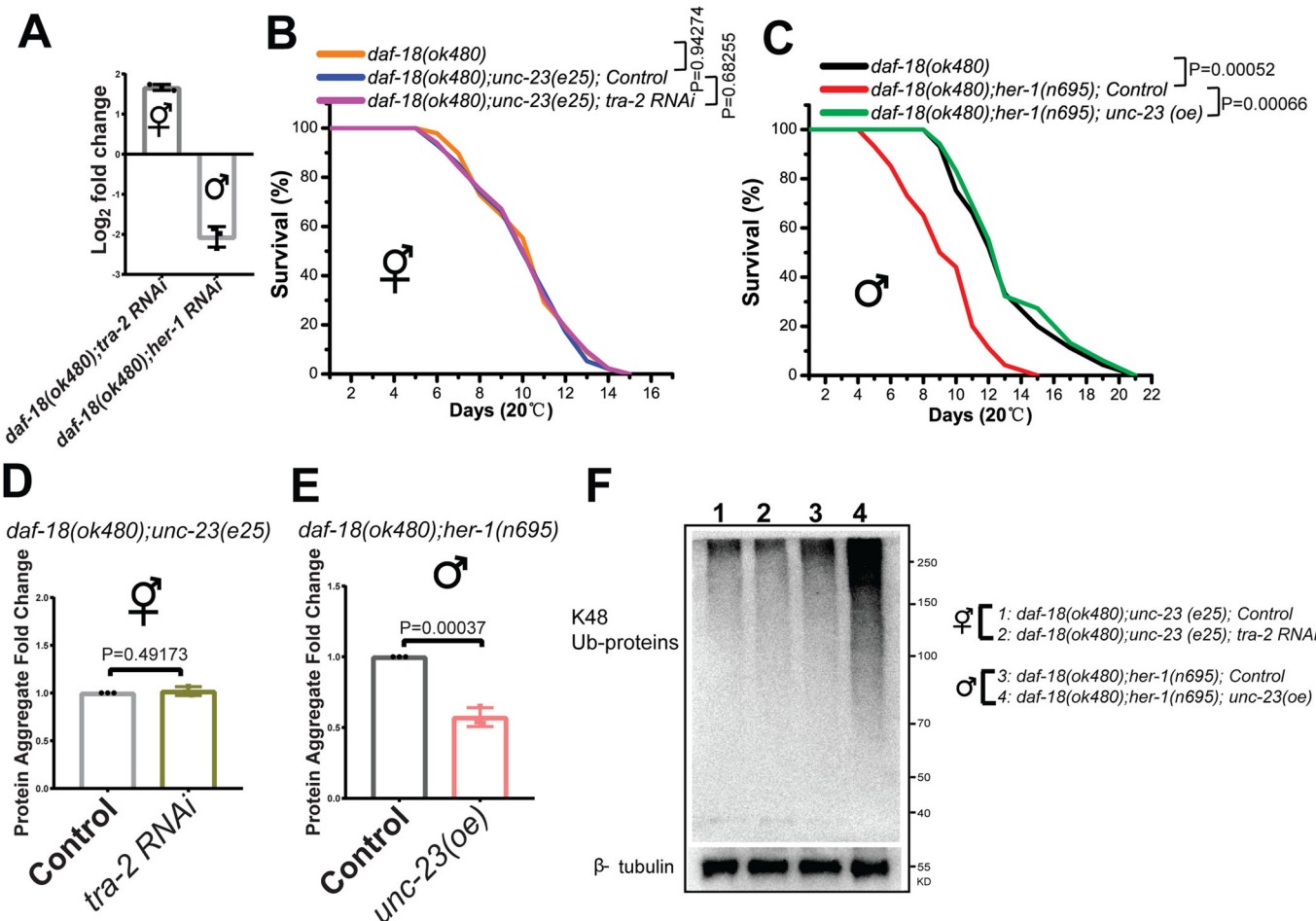

**Figure EV6.** *unc-23* **may be regulated by male sex determination pathway.**

(A) Real-time PCR tested the expression changes of *unc-23*. All these genes were significantly up- or downregulated by at least 2-fold. Log2FC: log2 (fold change) (>1 or <−1). The data show the average of three independent repeats, and the error bars show the standard deviations. (B) Enhancing male sex determination signaling by knocking down *tra-2* failed to extend the lifespan *daf-18(ok480);unc-23(e25)* hermaphrodites. Control: Feeding RNAi control clones with empty vector L4440. Sample size (n) = 96 (*daf-18*), 156 (*control*), 88 (*tra-2*). (C) Overexpressed *unc-23* in *daf-18(ok480); her-1(n695)* rescued the shorted lifespan. Control: the transgenic injection strains with empty expression vector L2528. Sample size (n) = 94 (*daf-18*), 72 (*control*), 63 (*unc-23 oe*). Each lifespan experiment set was repeated three times. The mean survival rates were calculated using the Kaplan–Meier method, and *P* values were determined by using the log rank test (*P* values; B, *daf-18(ok480)* hermaphrodites vs *daf-18(ok480);unc-23(e25)* hermaphrodites: 0.94274; *daf-18(ok480);unc-23(e25)* hermaphrodites vs *daf-18(ok480);unc-23(e25);tra-2 RNAi* hermaphrodites: 0.68255; C, *daf-18(ok480)* males vs *daf-18(ok480);her-1(n695)* males: 0.00052; *daf-18(ok480);her-1(n695)* males vs *daf-18(ok480);her-1(n695);unc-23(oe)* males: 0.00066). (D) The fold changes of protein aggregation in *daf-18(ok480);unc-23(e25)* hermaphrodites of three independent experiments when knocking down *tra-2*. Control: Feeding RNAi control clones with empty vector L4440. (E) The fold changes of protein aggregation in *daf-18 (ok480);her-1(n695)* males of three independent experiments when overexpressing *unc-23*. Control: the transgenic injection strains with empty expression vector L2528. The data show the average of three independent repeats, and the error bars show the standard deviations. *P* value was determined by using two-tailed *t-test* (*P* values; D, *control* hermaphrodites vs *tra-2 RNAi* hermaphrodites: 0.49173; E, *control* males vs *unc-23(oe)* males: 0.00037). (F) The k48 linked protein ubiquitination. Control for RNAi: Feeding RNAi control clones with empty vector L4440. Control for overexpression: the transgenic injection strains with empty expression vector L2528. All the lifespan data were summarized in Appendix Table S4. Each immunoblot shows representative of three times independent experiments, all the data details summarized in Appendix Table S5.

