## [Peer Review File · EMBO Reports]

Male sex determination maintains proteostasis and extends lifespan of *daf-18*/PTEN deficient *C. elegans*

Zhi Qu, Lu Zhang, Xue Yin, Fangzhou Dai, Wei Huang, Yutong Zhang, Dongyang Ran, and Shanqing Zheng

Corresponding author: Shanqing Zheng (zhengshanqing@henu.edu.cn)

Review Timeline:

Submission Date:	21st Aug 24
Editorial Decision:	8th Oct 24
Revision Received:	2nd Dec 24
Editorial Decision:	20th Dec 24
Revision Received:	24th Dec 24
Accepted:	7th Jan 25

Transaction Report:

Dear Prof. Zheng

Thank you for the submission of your research manuscript to our journal. I apologize for the delay in handling your manuscript, but we have now received the full set of referee reports that is copied below.

As you will see, the referees acknowledge that the findings are interesting and that the conclusions are overall supported by the data presented but they also raise a number of concerns and have suggestions how to further strengthen the data that will have to be addressed.

Given the constructive comments, we would like to invite you to revise your manuscript with the understanding that the referee concerns (as detailed above and in their reports) must be fully addressed and their suggestions taken on board. Please address all referee concerns in a complete point-by-point response. Acceptance of the manuscript will depend on a positive outcome of a second round of review. It is EMBO Reports policy to allow a single round of revision only and acceptance or rejection of the manuscript will therefore depend on the completeness of your responses included in the next, final version of the manuscript.

We realize that it is difficult to revise to a specific deadline. In the interest of protecting the conceptual advance provided by the work, we recommend a revision within 3 months (January 8th). Please discuss the revision progress ahead of this time with the editor if you require more time to complete the revisions.

I am also happy to discuss the revision further via e-mail or a video call, if you wish.

*******IMPORTANT NOTE:**

We perform an initial quality control of all revised manuscripts before re-review. Your manuscript will FAIL this control and the handling will be delayed IN CASE the following APPLIES:

- 1) A data availability section providing access to data deposited in public databases is missing. If you have not deposited any data, please add a sentence to the data availability section that explains that.
- 2) Your manuscript contains statistics and error bars based on $n=2$. Please use scatter blots in these cases. No statistics should be calculated if $n=2$.

When submitting your revised manuscript, please carefully review the instructions that follow below. Failure to include requested items will delay the evaluation of your revision. *****

- 1) a .docx formatted version of the manuscript text (including legends for main figures, EV figures and tables). Please make sure that the changes are highlighted to be clearly visible.
- 2) individual production quality figure files as .eps, .tif, .jpg (one file per figure). Please download our Figure Preparation Guidelines (figure preparation pdf) from our Author Guidelines pages <https://www.embopress.org/page/journal/14693178/authorguide> for more info on how to prepare your figures.
- 3) a .docx formatted letter INCLUDING the reviewers' reports and your detailed point-by-point responses to their comments. As part of the EMBO Press transparent editorial process, the point-by-point response is part of the Review Process File (RPF), which will be published alongside your paper.
- 4) a complete author checklist, which you can download from our author guidelines (<<https://www.embopress.org/page/journal/14693178/authorguide>>). Please insert information in the checklist that is also reflected in the manuscript. The completed author checklist will also be part of the RPF.
- 5) Please note that all corresponding authors are required to supply an ORCID ID for their name upon submission of a revised manuscript (<<https://orcid.org/>>). Please find instructions on how to link your ORCID ID to your account in our manuscript tracking system in our Author guidelines

(<<https://www.embopress.org/page/journal/14693178/authorguide#authorshipguidelines>>)

6) We replaced Supplementary Information with Expanded View (EV) Figures and Tables that are collapsible/expandable online. A maximum of 5 EV Figures can be typeset. EV Figures should be cited as 'Figure EV1, Figure EV2' etc... in the text and their respective legends should be included in the main text after the legends of regular figures.

7) Please include a dedicated "Data Availability" section at the end of the Methods (suggested wording: "The [structural coordinates | microarray | mass spectrometry] data from this publication have been deposited to the [name of the database] database [URL] and assigned the identifier [accession | permalink | hashtag]."). Should this not apply, this should still be stated as "This study includes no data deposited in external repositories."

See also < <https://www.embopress.org/page/journal/14693178/authorguide#dataavailability>>. Please note that the Data Availability Section is restricted to new primary data that are part of this study.

Additional information on source data and instruction on how to label the files are available <<https://www.embopress.org/page/journal/14693178/authorguide#sourcedata>>.

10) Figure legends and data quantification:

- the name of the statistical test used to generate error bars and P values,
- the number (n) of independent experiments (please specify technical or biological replicates) underlying each data point,
- the nature of the bars and error bars (s.d., s.e.m.)

- If the data are obtained from n {less than or equal to} 5, show the individual data points in addition to the SD or SEM.

- If the data are obtained from n {less than or equal to} 2, use scatter blots showing the individual data points.

11) Our journal encourages inclusion of *data citations in the reference list* to directly cite datasets that were re-used and obtained from public databases. Data citations in the article text are distinct from normal bibliographical citations and should directly link to the database records from which the data can be accessed. In the main text, data citations are formatted as follows: "Data ref: Smith et al, 2001" or "Data ref: NCBI Sequence Read Archive PRJNA342805, 2017". In the Reference list, data citations must be labeled with "[DATASET]". A data reference must provide the database name, accession number/identifiers and a resolvable link to the landing page from which the data can be accessed at the end of the reference. Further instructions are available at <<https://www.embopress.org/page/journal/14693178/authorguide#referencesformat>>.

12) All Materials and Methods need to be described in the main text using our 'Structured Methods' format. According to this format, the Methods section includes a Reagents and Tools Table (listing key reagents, experimental models, software and relevant equipment and including their sources and relevant identifiers) followed by a Methods and Protocols section describing the methods, ideally using a step-by-step protocol format. The aim is to facilitate adoption of the methodologies across labs. Please download and fill our Reagents and Tools Table template (.docx), which you can find in our author guidelines:

13) As part of the EMBO publication's Transparent Editorial Process, EMBO Reports publishes online a Review Process File to accompany accepted manuscripts. This File will be published in conjunction with your paper and will include the referee reports, your point-by-point response and all pertinent correspondence relating to the manuscript.

Yours sincerely,

=====

Referee #1:

The manuscript by Qu and colleagues focuses on the male survival advantage associated with daf-18/PTEN deficiency. Using the nematode *C. elegans* as a model and assessing recovery from L1 arrest and lifespan, Qu et al. show that, in contrast to wild type, daf-18 mutant males have a survival advantage over hermaphrodites. They performed an RNA sequencing analysis to identify genes that are differentially regulated in daf-18 mutant males and hermaphrodites. Their analysis revealed an enrichment of genes related to ER protein processing. Among them, they identified unc-23, whose expression is higher in daf-18 mutant males compared to hermaphrodites, as an important regulator of sex-specific survival differences. Depletion of unc-23 leads to UPRER induction and increased protein aggregation in both sexes. Interestingly, daf-18-deficient males (but not hermaphrodites) show increased K48 ubiquitination that is dependent on unc-23, whereas unc-23 overexpression per se is sufficient to increase K48 ubiquitination in hermaphrodites. This suggested that the survival advantage of daf-18-deficient males may be due to increased K48 ubiquitination and protein degradation (i.e., proficient proteostasis) as a result of inherently higher unc-23 expression. Next, the authors showed that the daf-18/PTEN protein phosphatase, but not its lipid phosphatase activity, is required for the survival advantage of males. Phosphoromics comparing WT and daf-18/PTEN protein phosphatase-deficient animals revealed that DAF-18 dephosphorylates a serine residue (S301) of the ER-localized protein C18E9.2/SEC-62. Increased phosphorylation of this residue led to increased ER stress and protein aggregation. However, daf-18 mutant males were able to circumvent the deleterious effects of aberrant C18E9.2 overphosphorylation by upregulating unc-23 expression and enhancing protein degradation. Finally, the authors show that daf-18 mutant males exhibit altered expression of sex determination pathway components, which in turn regulate unc-23 expression and increase proteasomal activity. This study is carefully planned, detailed and reports some very interesting findings. It paves the way for future studies to test whether similar mechanisms could explain the reduced cancer susceptibility of men with PTEN mutations compared to their female counterparts. Major comments

1) The inverse regulation of ER protein processing components in the two sexes of daf-18 mutants is intriguing. Is this differential expression pattern specific to daf-18 mutants or is it also evident when comparing wild-type males to hermaphrodites? This information should be provided as a heat map and/or gene expression analysis in the supplementary files.

2) The authors suggest that ER-associated protein degradation (ERAD) plays a central role in the survival advantage of daf-18 mutant males. Is this ER protein processing identified in their KEGG analysis identical to ER-associated degradation (ERAD)? For example, pek-1 is the PERK kinase homologue that is mainly involved in ERUPR induction. It should be tested whether knockdown of conserved ERAD components, such as SEL-1 or SEL-11, abolishes the extended lifespan of daf-18-deficient males.

3) In line with the previous comment, the expression of ERAD components is known to be regulated by the transcription factor XBP-1 (PMID: 34356855). It should be tested whether *xbp-1* mRNA splicing (or activation) differs between the two sexes in the *daf-18* mutant background.

4) The authors show that loss of *unc-23* (BAG2-type NEF) is associated with increased protein aggregation (Figure 3c, e) and decreased K48 ubiquitination levels (Figure 3h, j) in both sexes. Since *unc-23* itself is not a ubiquitin ligase, it is unclear what factors mediate these changes in ubiquitination status. Alternatively, decreased levels of ubiquitinated proteins could be due to increased proteasomal activity. It remains to be elucidated how these changes are mediated. UPS reporter assays combined with cycloheximide chase experiments and proteasomal inhibition could be helpful.

5) The authors suggest that constitutive phosphorylation in the S301 residue of the ER-resident protein C18E9.2 shortens the lifespan of hermaphrodites (Figure 5b). Could this be reversed by overexpression of *unc-23* (to mimic the "male" status)?

6) The observation that *daf-18* deficiency alters the expression of sex determination pathway components in males is very interesting but somewhat puzzling. DAF-18/PTEN is a phosphatase and not a transcription factor, so there is obviously an additional level of regulation for genes such as *her-1* and *tra-3* (Figure 6b-c).

Minor comments

1) Page 7, lines 183-185: The sentence: "Down-regulated *unc-23* in *daf-18(ok480)* hermaphrodites resulted in decreased attachment (Supplementary Figure 2), confirming that *daf-18*-deficient males up-regulated *unc-23* to properly attach their muscles" is complicated and potentially misleading. In Supplementary Figure 2, the authors show that muscle attachment to the hypodermis is more efficient in *daf-18* mutant males compared to hermaphrodites, presumably due to the elevated *unc-23* levels in males. The above sentence (in parentheses) could be simplified/reworded, perhaps as follows: "... suggesting that *unc-23*, which is up-regulated in males, helps them to attach their muscles more efficiently than hermaphrodites".

2) Page 9, line 245: The sentence: "However, the genomic *daf-18* can fully abolish the survival advantage of *daf-18* males" should be clarified. Figure 4e shows that overexpression of full-length *daf-18* under its endogenous promoter can completely abolish the survival advantage of *daf-18* mutant males.

3) It is necessary to use a consistent color code throughout the figures whenever possible. For example, in panels c and e of Figure 3, where the same genotypes are shown for the two different sexes.

Referee #2:

In this study Zhi Qu et al. used *C. elegans* to investigate dimorphism in the response to PTEN/*daf-18* deficiency. The authors found that male *C. elegans* are less affected by the deficits associated with *daf-18* protein phosphatase deficiency, which are mostly reflected in increased ER stress due to elevated phosphorylation of Sec62, which limits animal survival. Males overcome this stress by upregulating protein ubiquitination and degradation. This is achieved by elevating the levels of the UNC-23 E3 ligase in *daf-18(-)* males via the male sex determination pathway.

These findings are important as they pave the way towards understanding sexual dimorphism in health and disease, and whose underlying molecular mechanism is poorly understood. The work is very elaborate and includes both basic observations and mechanistic details. I find that the work presented here provides strong evidence for the existence and underlying mechanism for the sex-specific response to *daf-18* deficiency.

The experiments are clear and convincing. The major claims are novel and well supported. The findings are of interest to a wide variety of readers from the fields of aging, developmental biology, molecular biology and cell signaling fields.

That being said, my main problem with the manuscript is its organization. The introduction details too much of the experimental findings. Likewise, the discussion section contains experiments that should be in the results section! The methodology section is not detailed enough. I am also not sure the title grasps enough of the essence of the paper.

Introduction:

Page 3 line 89-Page 4 line 104 - this part doesn't belong to the introduction, at least not in this level of result details.

The sex determination pathway in *C. elegans* should be better introduced in the introduction.

Page 13 lines 363-410 should be moved to be immediately after the ubiquitination assays. And the MG132 experiment is very important. It should not be in the supp. (Supp fig 8)

Discussion

Page 17 lines 508-538 should all be incorporated in the result section

Methods

The methods are not detailed enough with too much reliance on previous descriptions in other papers. At least a short description of the procedure should be included in the text/

Survival and lifespan - were animals bleached?

RNAi protocol

Real time PCR - primers?

RNA sequencing - how many animals were collected? How were males and hermaphrodites separated?

Major comments:

Throughout the manuscript - the authors should better differentiate between L1 survival assays and adult lifespan assays. For

L1 arrest experiments clearly say that L1 arrested males live longer than L1 arrested hermaphrodites. For adult lifespan experiments clearly say that adult males live longer than adult hermaphrodites.
Likewise, throughout the manuscript, the authors should explicitly mention the genetic background (daf-18) of the worms when presenting the results. (for example in page 5 Lines 126, 127, 128.).
I believe the GO enriched category is ER related proteins (not ERAD)
Minor comments:
Page 3 line 75 - the statement that male worms normally have shorter lifespan than hermaphrodites is controversial, and depends on how the assay is setup.
Page 3 line 82 - why unfortunately?
Page 5 line 123 - "duration" is not the right term here
Page 5 - transcriptomics analysis. It should be emphasized that the comparisons were done within each sex, with or without daf-18.
Page 5 line 145 - "complicated" is not the formal wording to be used here.
Fig. 3g-j Protein ubiquitination quantification is needed
Page 8 lines 226,227 reduced degradation levels/protein degradation levels should be changed to Ubiquitination levels (which is what was tested directly).
Page 8 line 233 - loss of a phosphatase does not induce phosphorylation.
Page 8 line 234 - positive effects
Page 10 line 275 - dephosphorylation of C18E9.2 may be indirect (not necessarily by DAF-18 itself).
Fig. 5 title aggradation aggregation
Page 10 line 288 - Daf-18 should be daf-18
Page 11 line 305 - opposite to
Pag 12 line 340 - "function" should be changed to "expression"
Page 16 line 467 - how does your study shed light on the role of SEC62 in tumor biology?
Transgenic strain - better describe the cloning of unc-23 into the plasmid.
PROTEOSTAT analysis - what is the recommended concentration range?
26S fluorogenic assay - 25mg or ug?
Alpha synuclein - describe method in short
Protein phosphoromics - barely any details are provided.

Referee #1:

The manuscript by Qu and colleagues focuses on the male survival advantage associated with daf-18/PTEN deficiency. Using the nematode *C. elegans* as a model and assessing recovery from L1 arrest and lifespan, Qu et al. show that, in contrast to wild type, daf-18 mutant males have a survival advantage over hermaphrodites. They performed an RNA sequencing analysis to identify genes that are differentially regulated in daf-18 mutant males and hermaphrodites. Their analysis revealed an enrichment of genes related to ER protein processing. Among them, they identified unc-23, whose expression is higher in daf-18 mutant males compared to hermaphrodites, as an important regulator of sex-specific survival differences. Depletion of unc-23 leads to UPRER induction and increased protein aggregation in both sexes. Interestingly, daf-18-deficient males (but not hermaphrodites) show increased K48 ubiquitination that is dependent on unc-23, whereas unc-23 overexpression per se is sufficient to increase K48 ubiquitination in hermaphrodites. This suggested that the survival advantage of daf-18-deficient males may be due to increased K48 ubiquitination and protein degradation (i.e., proficient proteostasis) as a result of inherently higher unc-23 expression. Next, the authors showed that the daf-18/PTEN protein phosphatase, but not its lipid phosphatase activity, is required for the survival advantage of males. Phosphoromics comparing WT and daf-18/PTEN protein phosphatase-deficient animals revealed that DAF-18 dephosphorylates a serine residue (S301) of the ER-localized protein C18E9.2/SEC-62. Increased phosphorylation of this residue led to increased ER stress and protein aggregation. However, daf-18 mutant males were able to circumvent the deleterious effects of aberrant C18E9.2 overphosphorylation by upregulating unc-23 expression and enhancing protein degradation. Finally, the authors show that daf-18 mutant males exhibit altered expression of sex determination pathway components, which in turn regulate unc-23 expression and increase proteasomal activity.

This study is carefully planned, detailed and reports some very interesting findings. It paves the way for future studies to test whether similar mechanisms could explain the reduced cancer susceptibility of men with PTEN mutations compared to their female counterparts.

Major comments

1) The inverse regulation of ER protein processing components in the two sexes of daf-18 mutants is intriguing. Is this differential expression pattern specific to daf-18 mutants or is it also evident when comparing wild-type males to hermaphrodites? This information should be provided as a heat map and/or gene expression analysis in the supplementary files.

Response: Thank you for this suggestion. We used real-time PCR to analyze the gene expression levels of ER protein processing components unc-23, pek-1, skr-8, F44E5.4, and F44E5.5 in wild-type males and hermaphrodites. The results showed no significant differences in the expression of these genes,

suggesting that the expression changes may be specific to *daf-18* loss. Please see lines: 191-194 in main text and the added data in “Appendix Figure S4.” .

2) The authors suggest that ER-associated protein degradation (ERAD) plays a central role in the survival advantage of *daf-18* mutant males. Is this ER protein processing identified in their KEGG analysis identical to ER-associated degradation (ERAD)? For example, *pek-1* is the PERK kinase homologue that is mainly involved in ERUPR induction. It should be tested whether knockdown of conserved ERAD components, such as *SEL-1* or *SEL-11*, abolishes the extended lifespan of *daf-18*-deficient males.

Response: Thank you for this excellent suggestion. We found that the KEGG analysis and the genes summarized include the HSP70 complex, NEF, and PERK in the ER-related protein processing pathway, rather than the actual ERAD pathway. We tested *hsp-70* and the ERAD gene *pek-1*; however, knocking down *hsp-70* and *pek-1* did not significantly affect the survival of *daf-18* worms. We also followed your suggestion to test *sel-1* and *sel-11*, finding that *sel-11*, but not *sel-1*, significantly reduced but did not fully abolish the long lifespan of *daf-18* males. This suggests that the *sel-11*/HRD1 ERAD-L type ubiquitin ligase complex may be involved, but *unc-23* may also work with other ubiquitin ligases. We also updated the description from “ERAD-related pathway” to “ER-related protein processing” in the main text. Please see lines: 268-282 in main text, and the added data Figure EV4 in “EXPANDED VIEW FIGURES”.

3) In line with the previous comment, the expression of ERAD components is known to be regulated by the transcription factor XBP-1 (PMID: 34356855). It should be tested whether *xbp-1* mRNA splicing (or activation) differs between the two sexes in the *daf-18* mutant background.

Response: Thank you for this suggestion. We tested the *xbp-1* expression level in *daf-18* males and hermaphrodites and found a significant difference. XBP1 mRNA is induced by ATF6 and spliced by IRE1 in response to ER stress (Hiderou Yoshida et al., Cell 2001). Therefore, we knocked down *atf-6* and *ire-1* in worms to test if *xbp-1* is spliced and activated. We found that knocking down *atf-6*, *ire-1*, and *xbp-1* reduced the lifespan of *daf-18* males to some degree, though not as strongly as *unc-23*, suggesting that the *ire-1*/ERAD and *atf-6*/ERAD pathways do regulate the extended lifespan of *daf-18* males. However, the main factor may still be the *unc-23*-regulated Ub-proteasome degradation system. Please see lines: 268-282 in main text, and the added data Figure EV4 in “EXPANDED VIEW FIGURES”.

4) The authors show that loss of *unc-23* (BAG2-type NEF) is associated with increased protein aggregation (Figure 3c, e) and decreased K48 ubiquitination levels (Figure 3h, j) in both sexes. Since *unc-23* itself is not a ubiquitin ligase, it is unclear what factors mediate these changes in ubiquitination status. Alternatively, decreased levels of ubiquitinated proteins could be due to increased proteasomal activity. It remains to be elucidated how these changes are mediated. UPS reporter assays combined with cycloheximide chase experiments and proteasomal inhibition could be helpful.

Response: Thank you for this excellent suggestion. Since sel-11 is involved in the extended lifespan of daf-18 males, we believe that unc-23/NEF's unfolded protein selection may work with the ubiquitin ligase complex, including HRD1/sel-11, to influence ubiquitination. We once again sincerely thank you for this suggestion, which has made our work more complete, especially regarding the involvement of unc-23 in the ubiquitination process.

We also found that proteasomal activity is regulated by unc-23 and male sex determination genes, as shown by using a UPS activity reporter to monitor the degradation of specific fluorogenic peptide substrates (ZGly-Gly-Leu-AMC) and proteasomal inhibition assays (with the proteasome inhibitor MG132). Please see the data in Fig. 5B-C and Fig.9B-C. We also did lifespan, paralysis, a-beta aggregation, Motility, protein aggregation and alpha-synuclein inclusion analysis by using the proteasome inhibitor MG132. Please also see the data in Fig.5B-M, Fig. 8I-J and Fig. 9B-K.

If the data we showed not address your concerns, we sincerely look forward to further guidance.

5) The authors suggest that constitutive phosphorylation in the S301 residue of the ER-resident protein C18E9.2 shortens the lifespan of hermaphrodites (Figure 5b). Could this be reversed by overexpression of unc-23 (to mimic the "male" status)?

Response: Thank you for this excellent suggestion. We conducted the experiments according to your recommendation. We found that unc-23 overexpression could extend the lifespan of C18E9.2(S301E) hermaphrodites. Please see lines: 386-387 in main text and the added data Figure EV5 in "EXPANDED VIEW FIGURES".

6) The observation that daf-18 deficiency alters the expression of sex determination pathway components in males is very interesting but somewhat puzzling. DAF-18/PTEN is a phosphatase and not a transcription factor, so there is obviously an additional level of regulation for genes such as her-1 and tra-3 (Figure 6b-c).

Response: This is a very good and inspiring comment. We also noticed this and recently conducted a post-translational modification analysis, specifically focusing on protein phosphorylation in daf-18(ok480) and daf-18(D137A). We found that the loss of protein phosphatase activity in DAF-18 causes significant phosphorylation changes in two groups of proteins: the spliceosome (28 proteins) and the ribosome (20 proteins). This suggests that the protein phosphatase activity of DAF-18 may influence mRNA splicing and protein transcription. Consequently, DAF-18's phosphatase activity may affect her-1, tra-3 through these splicing and transcription processes. Identifying the direct targets of DAF-18's protein phosphatase activity is an ongoing project, and we hope to provide a clear answer to this question in the coming years.

Minor comments

1) Page 7, lines 183-185: The sentence: "Down- regulated unc-23 in daf-18(ok480) hermaphrodites resulted in decreased attachment (Supplementary Figure 2), confirming that daf-18-deficient males up-regulated unc-23 to properly attach their muscles" is complicated and potentially misleading. In Supplementary Figure 2, the authors show that muscle attachment to the hypodermis is more efficient in daf-18 mutant males compared to hermaphrodites, presumably due to the elevated unc-23 levels in males. The above sentence (in parentheses) could be simplified/reworded, perhaps as follows: "... suggesting that unc-23, which is up-regulated in males, helps them to attach their muscles more efficiently than hermaphrodites".

Response: Thank you very much for this suggestion, and we modified according to your suggestion. Please see lines: 223-224 in main text.

2) Page 9, line 245: The sentence: "However, the genomic daf-18 can fully abolish the survival advantage of daf-18 males" should be clarified. Figure 4e shows that overexpression of full-length daf-18 under its endogenous promoter can completely abolish the survival advantage of daf-18 mutant males.

Response: Thank you very much for this suggestion, and we modified according to your suggestion. Please see lines: 304-306.

3) It is necessary to use a consistent color code throughout the figures whenever possible. For example, in panels c and e of Figure 3, where the same genotypes are shown for the two different sexes.

Response: We tried to keep the colors consistent for the same genotype within each experiment in each figure. Please see all the modified figures.

Referee #2:

In this study Zhi Qu et al. used *C. elegans* to investigate dimorphism in the response to PTEN/*daf-18* deficiency. The authors found that male *C. elegans* are less suffer from the deficits associated with *daf-18* protein phosphatase deficiency, which are mostly reflected in increased ER stress due to elevated phosphorylation of Sec62, which limits animal survival. Males overcome this stress by upregulating protein ubiquitination and degradation. This is achieved by elevating the levels of the UNC-23 E3 ligase in *daf-18(-)* males via the male sex determination pathway.

These findings are important as they pave the way towards understanding sexual dimorphism in health and disease, and whose underlying molecular mechanism is poorly understood. The work is very elaborate and includes both basic observations and mechanistic details. I find that the work presented here provides strong evidence for the existence and underlying mechanism for the sex-specific response to *daf-18* deficiency.

The experiments are clear and convincing. The major claims are novel and well supported. The findings are of interest to a wide variety of readers from the fields of aging, developmental biology, molecular biology and cell signaling fields.

That being said, my main problem with the manuscript is its organization. The introduction details too much of the experimental findings. Likewise, the discussion section contains experiments that should be in the results section! The methodology section is not detailed enough.

I am also not sure the title grasps enough of the essence of the paper.

Response: we changed the details according to all your suggestions and the title is changed as "Male Sex Determination Improves Survival Outcomes in the Absence of *daf-18*/PTEN via *unc-23*/NEF".

Introduction:

Page 3 line 89-Page 4 line 104 - this part doesn't belong to the introduction, at least not in this level of result details.

Response: We accept your suggestion and have deleted some details, focusing only on several important points. Please see the following lines:111-124.

The sex determination pathway in *C. elegans* should be better introduced in the introduction.

Response: We accept your wonderful suggestion and have added more details about sex determination in the introduction. Please see the following lines:84-104.

Page 13 lines 363-410 should be moved to be immediately after the

ubiquitination assays. And the MG132 experiment is very important. It should not be in the supp. (Supp fig 8)

Response: Since the data in lines 363-410 were primarily obtained using daf-18(D137A), after fully considering your suggestion and the presented data, we moved this information to follow the ubiquitination assays and daf-18(D137A) lifespan assays. Please see the following lines: 323-372. Additionally, the MG132 experiment data has been moved to the main text accordingly; please see the Fig. 5L-M and Fig. 8I-J.

Discussion

Page 17 lines 508-538 should all be incorporated in the result section

Response: We accept your suggestion, and all of this data has now been incorporated into the results section. Please see the following lines:148-168.

Methods

The methods are not detailed enough with too much reliance on previous descriptions in other papers. At least a short description of the procedure should be included in the text/

Survival and lifespan - were animal bleached?

Response: The experimental details have been added. Please see lines:610-613.

RNAi protocol

Response: The protocol has been added. Please see line:628-637.

Real time PCR - primers?

Response: All PCR primers were summarized in Table S6(Appendix).

RNA sequencing - how many animals were collected? How were males and hermaphrodites separated?

Response: The males were separated by picking, and this process needs to be completed at young L4 stage to avoid mating. The time window for this process is narrow, and we required a large number of males to prepare each sample. The process was time-consuming; we had five people picking for several hours to prepare just one sample.

Typically, 50-100 mg of worm pellets were used for RNA sequencing, although fewer worms could be picked as long as the total RNA yield passed the quantification test (about 50 ng/ μ L, 35 μ L in total) of the sequencing company.

The details were added now, please see lines: 665-671.

Major comments:

Throughout the manuscript - the authors should better differentiate between L1 survival assays and adult lifespan assays. For L1 arrest experiments clearly say that L1 arrested males live longer than L1 arrested hermaphrodites. For adult lifespan experiments clearly say that adult males live longer than adult hermaphrodites.

Response: We accepted your suggestion, and all survival tests have been clearly described.

Likewise, throughout the manuscript, the authors should explicitly mention the genetic background (daf-18) of the worms when presenting the results. (for example in page 5 Lines 126, 127, 128.).

Response: We accepted your suggestion. The alleles of all these worms have been added.

I believe the GO enriched category is ER related proteins (not ERAD)

Response: We agree and changed to "ER-related protein processing".

Minor comments:

Page 3 line 75 - the statement that male worms normally have shorter lifespan than hermaphrodites is controversial, and depends on how the assay is setup.

Response: We deleted the statement.

Page 3 line 82 - why unfortunately?

Response: We deleted the word "unfortunately".

Page 5 line 123 - "duration" is not the right term here

Response: We changed it to survival. please see line 133.

Page 5 - transcriptomics analysis. It should be emphasized that the comparisons were done within each sex, with or without daf-18.

Response: We emphasized this. please see lines:174-176.

Page 5 line 145 -"complicated" is not the formal wording to be used here.

Response: We changed it to "enormous". please see line 178.

Fig. 3g-j Protein ubiquitination quantification is needed

Response: All the ubiquitination experiments were quantified and summarized, please see Fig. 3G-J.

Page 8 lines 226,227 reduced degradation levels/protein degradation levels should be changed to Ubiquitination levels (which is what was tested directly).

Response: We changed according to your suggestion, please see line 284-285.

Page 8 line 233 - loss of a phosphatase does not induce phosphorylation.

Response: We changed it to "Loss of daf-18 inactivates the IIS transcription factors DAF-16....." . Please see line 291.

Page 8 line 234 - positive effects

Response: We modified to "positive effects" according to your suggestion, please see line 293.

Page 10 line 275 - dephosphorylation of C18E9.2 may be indirect (not necessarily by DAF-18 itself).

Response: We changed in to "DAF-18 directly or indirectly dephosphorylates C18E9.2". please see: line 381.

Fig. 5 title aggradation↯aggregation

Response: We modified the typo.

Page 10 line 288 - Daf-18 should be daf-18

Response: We modified it according to your suggestion.

Page 11 line 305 - opposite to

Response: We modified it according to your suggestion. Please see line 420.

Page 12 line 340 - "function" should be changed to "expression"

Response: we modified it according to your suggestion. Please see line 459.
Page 16 line 467 - how does your study shed light on the role of SEC62 in tumor biology?

Response: C18E9.2 is predicated to be a homology of human SEC62, we described this. please see line 532.

And we changed it to "Our study may shed light on the role of SEC62 in aging biology." Please see line 550.

Transgenic strain - better describe the cloning of unc-23 into the plasmid.

Response: We added some details about the cloning of unc-23, please see lines 679-689.

PROTEOSTAT analysis - what is the recommended concentration range?

Response: 1 µg/ml to 10 mg/ml, this has been added in the method. Please see lines 700-701.

26S fluorogenic assay - 25mg or ug?

Response: Should be 25 µg, we modified it. Please see line 718.

Alpha synuclein - describe method in short

Response: We describe the method, please see lines: 726-733.

Protein phosphoromics - barely any details are provided.

Response: The method detail now is provided, please see lines: 748-757.

Dear Prof. Zheng

Thank you for the submission of your revised manuscript to EMBO reports. We have now received the report from the referee who was asked to assess it. As you will see, s/he is very positive about the study and supports publication.

Browsing through the manuscript myself, I noticed a few editorial things that we need before we can proceed with the official acceptance of your study.

- The Data Availability section needs to go before the Acknowledgments. It is restricted to referring to data deposited in external repositories. Therefore, please remove the statement "All data are available in the main text and the EXPANDED VIEW RESULTS."

- We further need the specific URL for GSE222447 dataset in the data availability statement.

- In checking the methods section, I noticed two typos. Line 610: "In brief, Well-maintained..." should be "In brief, well-maintained...". Line 611: you mention "blench buffer" - should this be "bleach buffer"?

- Regarding the Author Contributions, we now use CRediT to specify the contributions of each author in the journal submission system. CRediT replaces the author contribution section, which therefore needs to be removed from the manuscript text. You can use the free text box in our system if you wish to provide more detailed descriptions. See also guide to authors <https://www.embopress.org/page/journal/14693178/authorguide#authorshipguidelines>.

- Please add callouts for Appendix Tables S2, S3 and S4, e.g., when you callout the related Appendix Figure.

- Please remove the synopsis text from the manuscript file and upload it as a separate word file. For display on our homepage, the text needs to be shortened to a 1-2 sentence summary and 2-3 bulletpoints.

If you wish, you could add a model/summary figure to the manuscript and then keep the longer text as figure legend. (Please change "ER member-localized protein" to "ER membrane-localized" - auto-correction error, I assume).

- Our production/data editors have asked you to clarify several points in the figure legends (see below). Please incorporate these changes in the manuscript and return the revised file with tracked changes with your final manuscript submission.

A) Statistical test information. Only p-values that are actually shown in the figure panel(s) should (and must) be defined in the legends, all others should be removed from (or added to) the legend. Moreover, we ask for the specification of exact p-values:

- Please note that the exact p values are not provided in the legends of figures 1C, F; 5B, C, D, E, H, I, J, K; 6D, E, F, H; 7K, 8E, J; 9E, I, K; EV4B.

- Please indicate the statistical test used for data analysis in the legends of figures 3A, B, D, F, G, H, I, J; EV3 B.

B) Replicates and error bars:

- Please note that information related to n is missing in the legends of figures 2E, 5A-E; 7C, 8A.

- Please note that the error bars are not defined in the legends of figures 1A, B; 2E; 5A-E, H, I, L, M; 6F, G, H, I; 7C, 9A, B, C, D, E, H, I, J, K; EV2, EV3 A, B; EV4 A; EV6A, D, E.

- Please note that the measure of center for the error bars needs to be defined in the legends of figures 2F, G; 3A, B, D, F, G, H, I, J; 5J, K; 7D-G, L-O; 8A, C, D, F, G.

C) Data presentation:

- Please note that the scale bar needs to be defined for figures 7L, N

- Appendix Figure legends

Appendix Figure S3 is missing information on 'n' in the legend.

Appendix Figure S4: Please define the bars and error bars as well as 'n'

Appendix Figure S6: please define the bars and error bars in the legend and whether the data are representative of one independent repeat or include all three repeats. Please define the statistical test used and please only calculate p-values if the data are a comparison across all 3 independent repeats, but not one repeat with x # of worms.

- Appendix Figure S7: it is very difficult to read the GO terms and upon zooming it, the resolution of the text is not that great. What about splitting the data shown in panel A into three panels, e.g. one showing GO terms "biological process", one showing the GO terms "cellular component" etc.

- Appendix Figure S8 and S11: please define bars and error bars.

- Once resized to the final size of 550 pixels width, the synopsis image could do with a somewhat larger font size of the labels in the top-left part (e.g., DAF-18/PTEN protein phosphatase is a bit difficult to read).

- Source data: if possible, the source data need to be re-organized so that all panels of one figure are in one folder. Then the zipped folder for each figure is uploaded as e.g., Figure 1 Source Data, Figure 2 Source Data etc. Moreover, each figure folder should have separate files/subfolders for each panel. Please also note the nomenclature 'source data' and correct the typo - likely the autocorrection function - 'sauce data'.

- As a standard procedure we edit abstract and title to make them more accessible to our general readership. Please find my suggestions below my signature.

With kind regards,

=====

Referee #1:

The revised version of the manuscript addressed all remaining questions/points and will be of interest to the readership of EMBO reports.

=====

Male sex determination maintains proteostasis and extends lifespan of daf-18/PTEN deficient *C. elegans*

Although females typically have a survival advantage, those with PTEN functional abnormalities face a higher risk of developing tumors than males. However, the differences in how each sex responds to PTEN dysfunction have rarely been studied. We use *Caenorhabditis elegans* to investigate how male and hermaphrodite worms respond to dysfunction of the PTEN homolog daf-18. Our study reveals that male worms can counterbalance the negative effects of daf-18 deficiency, resulting in longer adult lifespan. The survival advantage depends on the loss of DAF-18 protein phosphatase activity, while its lipid phosphatase activity is dispensable. The deficiency in DAF-18 protein phosphatase activity leads to the failure of dephosphorylation of the endoplasmic reticulum membrane protein C18E9.2/Sec62, causing increased levels of unfolded and aggregated proteins in hermaphrodites. In contrast, males maintain proteostasis through a UNC-23/NEF-mediated protein ubiquitination and degradation process, providing them with a survival advantage. We find that sex determination is a key factor in regulating the differential expression of unc-23 between sexes in response to daf-18 loss. These findings highlight the unique role of the male sex determination pathway in regulating protein degradation.

The authors addressed the remaining editorial issues.

Prof. Shanqing Zheng
Henan University
1 Jinming Avenue North, Longting District, School of Basic Medical Sciences, Henan University
Kaifeng 475004
China

Dear Shanqing,

I am very pleased to accept your manuscript for publication in the next available issue of EMBO reports. Thank you for your contribution to our journal.

Kind regards,

Martina
